# Electrifying long-haul freight trucks reduces societal costs in the United States

Jason Porzio [1,2], Wilson McNeil[1,2], Fan Tong [3,4,5], Scott Moura [2], Maximilian Auffhammer [6,7] ✉ & Corinne D. Scown [1] ✉

Electrifying long-haul heavy-duty vehicles (HDVs) entails high private costs but offers substantial reductions in external costs by substituting diesel combustion with electricity generation. We combine technoeconomic analysis and life-cycle assessment of lithium-ion battery electric (BE) and diesel HDVs to estimate total private costs and monetized climate and health damages in the United States. In 2025, BE-HDVs are estimated to have 46% higher private costs ($0.71 mile$^{-1}$) than diesel trucks, decreasing to 33% ($0.52 mile$^{-1}$) by 2035. However, their external costs are 64–69% lower in 2025 and 70–80% lower in 2035. Overall, BE-HDVs yield positive net societal benefits by 2035, contingent on policies that accelerate their adoption.

The transportation sector is the largest emitter of greenhouse gases (GHGs) in the US, accounting for 29% of domestic GHG emissions in 2022[1]. Heavy-duty vehicles (HDVs) contribute a disproportionate share, accounting for 27% of on-road GHG emissions – despite representing 1% of on-road vehicles – and 15% of total transportation emissions[2,3]. The use of diesel in HDV internal combustion engines also contributes 50% of fine particulate matter (PM$_{2.5}$) emissions from on-road vehicles and their associated human health burdens, often in highly populated urban corridors, which is linked to disproportionate negative impacts to disadvantaged communities[3–7]. While there are several mandates and legislation at the state and national levels to electrify, decarbonize, and diminish the human health impacts of HDVs, the technological pathway offering the highest net social benefits remains uncertain[8–10].

Lithium-ion (Li-ion) batteries are a leading candidate for electrifying HDVs, largely due to the rapidly growing popularity of Li-ion battery passenger electric vehicles (EVs) and their declining prices[11]. Since 2010, the size of the US passenger EV fleet has increased by nearly two orders of magnitude, in part due to the nearly 90% decrease in Li-ion battery prices[12,13]. Despite this, few commercial options exist for Li-ion battery electric HDVs (BE-HDVs), particularly for long-haul freight, defined as trips over 250 miles (Fig. S2). Since long-haul freight contributes 68% of GHG emissions from HDVs[2], decarbonization of HDVs is not achievable without solutions for long-haul trucks. BE-HDVs

offer potential reductions to GHG and air pollutant emissions, but their high total cost of ownership (TCO) poses a barrier to adoption. Current literature agrees that the TCO of BE-HDVs is at least $0.40 per vehicle mile traveled (VMT) higher than diesel HDVs[14,15], yet a definitive answer to the balance of costs and benefits of BE-HDVs is elusive.

This study provides a comprehensive comparison of the private and external costs of long-haul BE-HDVs relative to diesel. We achieved this by performing a technoeconomic analysis and life-cycle assessment of BE-HDVs and diesel HDVs (Fig. 1). Included in this, we compiled upfront material and manufacturing impacts, modeled long-haul HDV and battery charging behavior across the US, simulated individual electricity generator responses to location-specific changes in charging loads, calculated emissions from future induced electricity generation and diesel combustion, and modeled the private and external costs attributable to operations of BE-HDVs and diesel HDVs. Specifically, we evaluate whether the electrification of marginal long-haul HDVs with Li-ion batteries is projected to reduce GHG emissions and decrease the burden on human health, and how the value of those impacts compares to the private costs for truck operators. We further explore the conditions required to make long-haul BE-HDVs economically and socially attractive from a life-cycle perspective. To answer this, we compare the private and external costs (e.g., health damages from local pollutants and global damages from GHG emissions) of long-haul BE-HDVs and diesel HDVs

[1]Energy and Biosciences Institute, University of California, Berkeley, Berkeley, CA, USA. [2]Civil and Environmental Engineering Department, University of California, Berkeley, Berkeley, CA, USA. [3]School of Economics and Management, Beihang University, Beijing, People's Republic of China. [4]Lab for Low-carbon Intelligent Governance, Beihang University, Beijing, People's Republic of China. [5]Peking University Ordos Research Institute of Energy, Ordos City, Inner Mongolia, People's Republic of China. [6]Department of Agricultural and Resource Economics, University of California, Berkeley, Berkeley, CA, USA. [7]National Bureau of Economic Research, Cambridge, MA, USA. ✉e-mail: aufhammer@berkeley.edu; cscown@berkeley.edu

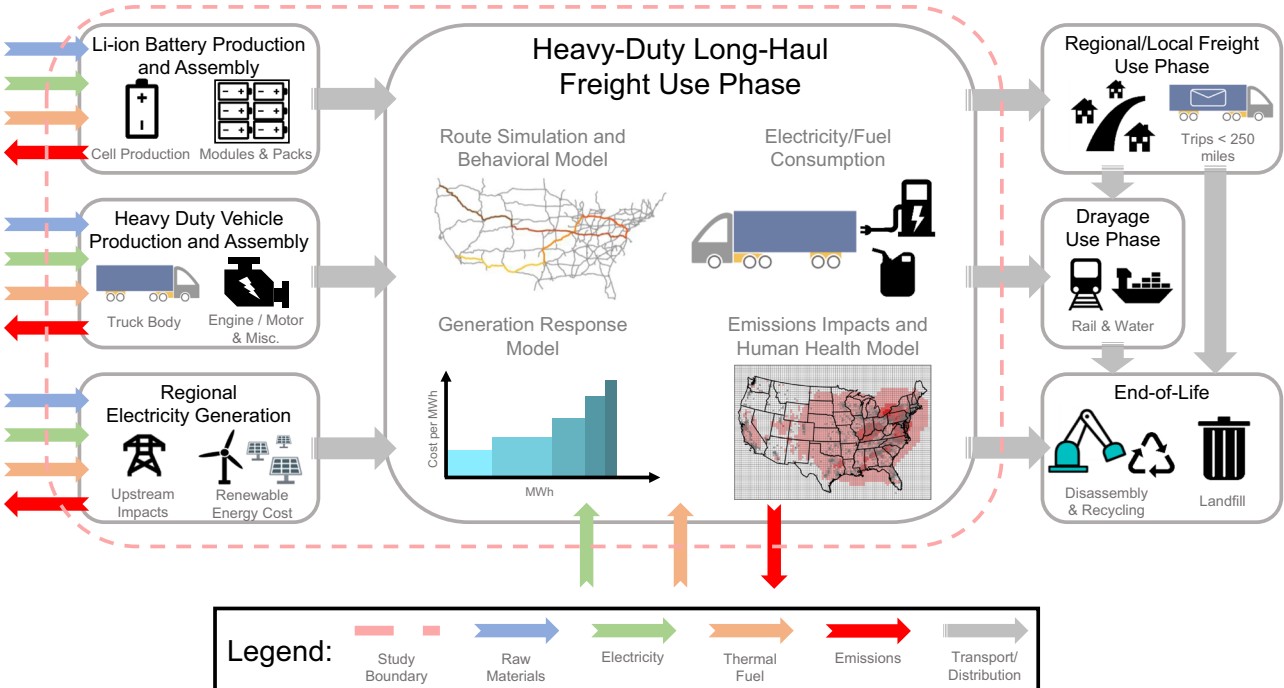

**Fig. 1 | Schematic representation of life-cycle assessment and technoeconomic analysis for long-haul BE-HDVs and diesel HDVs.** Overview of the life-cycle assessment and technoeconomic analysis used to simulate the private and external costs of long-haul BE-HDVs and diesel HDVs in the United States. Geographic data on contiguous United States boundaries and truck corridors is made available by the United States Census Bureau[93] and the United States Bureau of Transportation Statistics[70] respectively.

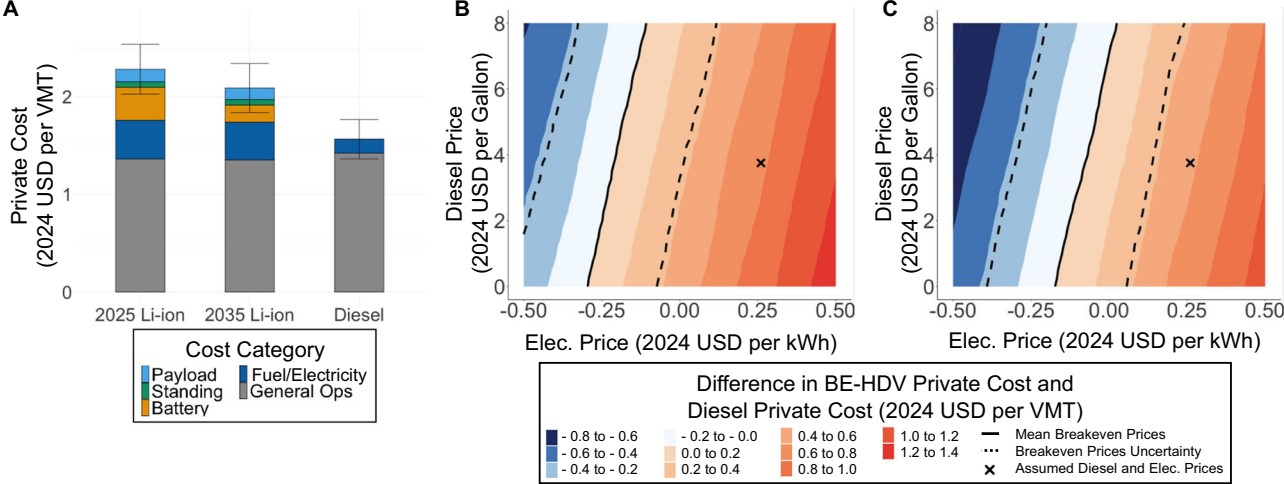

**Fig. 2 | Private NPVs and sensitivity for long-haul BE-HDVs and diesel HDVs in varying years. A** Private cost per VMT of long-haul BE-HDVs in 2025 and 2035 compared to diesel HDVs. Private costs are determined through NPV analysis. Uncertainty bars represent two standard deviations. **B** Sensitivity of the difference in BE-HDV private costs and diesel private costs to electricity and diesel prices in 2025. **C** Sensitivity of the difference in BE-HDV private costs and diesel private costs to electricity and diesel prices in 2035. Uncertainty bars represents two standard deviations on BE-HDV and diesel HDV private costs higher and lower.

under low and high renewable energy cost scenarios in 2025 and 2035 while accounting for changes in battery technology and electricity generation. Additionally, we perform a sensitivity analysis to describe how the private costs of long-haul BE-HDV's vary relative to diesel HDVs given different electricity and diesel prices.

## Results
### Private costs
We estimate the private costs of marginal Li-ion BE-HDVs and diesel HDVs by modeling the annual activity of a fleet of 100 HDVs performing long-haul trips via a Markov Chain Monte Carlo simulation and determining their average net present value (NPV) over their four-year average lifetime[14–16]. Further elaboration on this modeling decision is provided in Discussion S2. Figure 2 summarizes these resulting private cost per VMT and the sensitivity of private costs to diesel prices and charging prices for BE-HDVs with NMC811 Li-ion batteries, abbreviated to "Li-ion." Figures S3–S13 visualize these results for all Li-ion battery chemistries examined (NMC811, NCA, and LFP) and discount rates. We model BE-HDVs and diesel HDVs as class 8 trucks since these vehicles made up over 80% of new HDV truck sales as of 2021[3].

Table S3 provides truck characteristics including battery capacity and range. Additional details on the simulation of BE-HDVs and diesel HDVs are provided in the Methods section. Table S12 provides average US electricity generation marginal emission factors excluding solar and wind generation. Table S13 provides the average US marginal emissions per total electricity demand from battery charging by scenario.

Private costs consist of five categories. "General Operations (Ops)" includes costs of vehicle depreciation, insurance, taxes, additional fees, maintenance, and driving labor. "Fuel/Electricity" represents costs of diesel or electricity to refuel or recharge the HDV. "Battery" represents the depreciation of the battery while performing long-haul freight transport. We modeled the battery to reach end-of-life and achieve full depreciation after 4 years, the length of the typical long-haul freight use-phase for HDVs[14-16]. "Standing" includes additional time and labor costs attributable to charging the battery. "Payload" represents the cost of lost weight capacity available for cargo due to the additional battery weight. In Fig. 2A, the charging electricity price is fixed at $0.26/kWh for all years to mitigate its high uncertainty. This value includes the price of utilizing charging infrastructure set at $0.09/kWh as estimated by Burnham et al.[17] and the price of energy delivered to BE-HDVs set at $0.17/kWh to agree with US EIA's[18] electricity price estimate for transportation. The price of diesel is set at $3.75/gal[18]. The sensitivity analysis in Fig. 2B and C are performed to capture regional and temporal variations in diesel and electricity prices, as well as uncertainty in costs of establishing a nationwide fast-charging network. We employ a 17.3% learning rate to forecast Li-ion battery prices[19] and use a 2% discount rate when calculating the NPV.

As visualized in Fig. 2A, the current private costs of diesel HDVs are substantially lower than BE-HDVs. The uncertainty bars convey the impacts of truck model parameter ranges, battery characteristics, and variations in trucking behavior. The private costs of diesel HDVs in 2025 is $1.57/VMT, while the private costs of BE-HDVs are 46% higher at $2.28/VMT (18% to 79% with uncertainty).

The private costs of BE-HDVs in 2035 are projected to be 8% lower than in 2025, driven by declining battery prices. However, barring unexpected technological or economic changes, the private cost of diesel HDVs remains lower than those of BE-HDVs. The private cost of long-haul diesel HDVs in 2035 remains at $1.57/VMT, assuming that diesel HDV design remains relatively constant and technological improvements result in incremental performance changes[20,21]. The private costs of BE-HDVs decrease to $2.09/VMT, 34% higher than diesel (8% to 65% with uncertainty). This decrease is entirely attributable to battery prices, which assumedly decrease by 48% from 2025 to 2035 under a 17.3% learning rate and the global demand forecasted by BloombergNEF[13].

Across all BE-HDV scenarios, electricity represents the largest cost after General Ops. However, there is substantial uncertainty regarding electricity prices at charging stations across regions and time given the current lack of widespread charging infrastructure for high-capacity BE-HDVs along trucking corridors and the possibility of electricity rate structure changes. Additionally, while fuel only makes up 9% of the private costs of diesel HDVs, diesel prices remain uncertain considering future economic scenarios and regional variation. Figure 2B and C explore the impact of different electricity and diesel prices on the difference in private cost per VMT of BE-HDVs and diesel HDVs. Additionally, Figs. S14 and S15 illustrate the social costs under a variety of electricity prices.

In 2025, the breakeven point of BE-HDVs, defined as the electricity and diesel prices where private costs of diesel HDVs and BE-HDVs are equivalent, is -$0.21/kWh (-$0.44/kWh to $0.01/kWh with uncertainty) given the default diesel price. In 2035, this breakeven electricity price of BE-HDVs reaches -$0.08/kWh (- $0.30/kWh to $0.14/kWh with uncertainty). The upper range of uncertainty remains below likely retail prices and is dependent on changes in truck parameters and behavior that drive uncertainty. Additionally, this outcome depends on future battery prices, the driver of reduced BE-HDV private costs by 2035.

Even when considering the highest modeled diesel prices of $8.00/gal, negative electricity prices are still needed to achieve a breakeven point in both 2025 and 2035. However, breakeven points with positive electricity prices are possible at the upper range of uncertainty. This occurs with diesel prices greater than $3.31 in 2025 and for all diesel prices in 2035. However, even at this upper model boundary, the electricity price will always need to be lower than the current electricity prices modeled for all diesel prices studied. Therefore, substantial changes to diesel and electricity prices are required to lower the private costs of long-haul BE-HDVs below diesel HDVs, even when considering future battery prices.

## External costs

Emissions from simulated truck behavior are modeled and used to characterize the external costs of BE-HDVs and diesel HDVs. Figure 3A and 3B visualize $PM_{2.5}$ emissions in 2025 under Low RE from the simulated diesel HDV and BE-HDV behavior respectively, while Fig. 3C presents the external NPV per VMT. Results are presented for high and low renewable energy cost scenarios (High RE and Low RE, respectively) which affect what regional generator type is on the margin (i.e. comes online or ramps up to meet new demand). The SCCs of $212/tonne $CO_{2eq}$ and $248/tonne $CO_{2eq}$ are used to determine damages associated with GWP (using the Intergovernmental Panel on Climate Change's 100-year values in the Assessment Report 6) in the 2025 and 2035 scenarios respectively (SCC's visualized in Fig. S16)[22]. A 2% discount rate is used for the "Human Health" category and all GWP categories in accordance with the EPA's SCC calculation from the Office of Management and Budget guidance[22] and standard practice on social discounting[23]. Human health impacts included in this study are limited to long-haul HDV tailpipe emissions and electricity generator stack emissions due to the limited availability and high uncertainty of modeling human health damages from battery manufacturing and material extraction, which typically occurs outside the US. Even for impacts within the US, it is worth noting that the translation to monetized damages through the use of SCC and value of a statistical life (VSL) come with substantial uncertainty, and a full exploration of this uncertainty warrants additional studies dedicated to this topic. We apply these costs in a consistent manner across BE-HDV and diesel HDV scenarios to ensure a fair comparison. However, uncertainty associated with the SCC and VSL can impact the relative importance of private and external costs in the total social cost calculations discussed in the following section. Results for additional battery chemistries and discount rates are presented in Figs. S17–S21.

External costs are grouped into seven categories. "GWP Truck" includes GWP from emissions related to the material extraction and manufacturing of truck components excluding the battery, which are separately reported in the "GWP Battery" category. "GWP Diesel Upstream" includes upstream GHG emissions from crude oil extraction, transportation to refineries, refining, and finished product transportation. "GWP Elec. Upstream" represents upstream GHG emissions attributable to the manufacturing of infrastructure supporting electricity generation and the extraction/transportation of fuels based on the average US grid mix[22]. "GWP Diesel Combustion" includes GWP impacts attributable to direct combustion of diesel. "GWP Elec. Gen. and Loss" represents GWP attributable to electricity generation, transmission line losses, and losses from battery charging. "Human Health" includes monetized damages to human health, specifically premature mortality modeled using InMAP[24,25] from the emissions of local air pollutants (e.g., $PM_{2.5}$, $SO_2$, $NO_x$) from diesel combustion and electricity generation. Additional details are provided in the Methods section.

In all scenarios, the average external costs (based on human health and GWP) from long-haul BE-HDVs nationwide are substantially

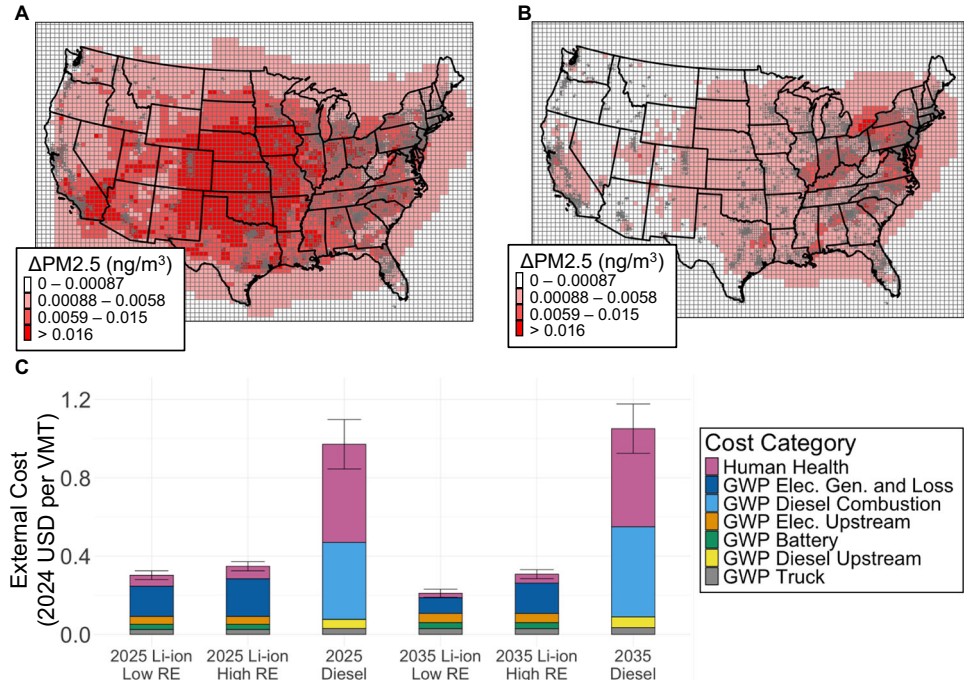

**Fig. 3 | External costs of long-haul BE-HDVs and diesel HDVs in varying years and renewable energy cost scenarios. A** Change in $PM_{2.5}$ from the simulated marginal fleet of 100 diesel HDVs in 2025. **B** Change in $PM_{2.5}$ from the simulated marginal fleet of 100 BE-HDVs in 2025 under a low renewable energy cost scenario. **C** Visualized external costs per VMT. External costs are determined through NPV analysis. Uncertainty bars represent two standard deviations. Geographic data for the map of the contiguous United States boundaries is made available by the United States Census Bureau[93].

lower than the external costs from diesel. The difference between BE-HDVs and diesel HDVs is most stark for human health impacts (Fig. 3C). In 2025 under Low RE, the human health damages for BE-HDVs are 89% lower (81-93%, with uncertainty) than diesel HDVs, with the difference increasing to 95% (92-97%, with uncertainty) in 2035. The total average external cost of BE-HDVs is $0.30/VMT, which is 69% (61-74%, with uncertainty) lower than diesel HDVs at $0.97/VMT. In 2025 under High RE, which relies more heavily on thermal generators on the margin, the average external cost of BE-HDVs is $0.35/VMT, 64% (56-70%, with uncertainty) lower than diesel HDVs.

By 2035, the relative advantage of BE-HDVs increases from an external cost perspective. Under Low RE, the average external costs for BE-HDVs are $0.21/VMT, 80% (75-84% with uncertainty) lower than diesel HDVs at $1.05/VMT, which is higher than its 2025 external cost due to increasing SCCs in accordance with the EPA's practices[22]. The High RE scenario still results in substantial reductions in external costs for BE-HDVs relative to diesel ($0.31/VMT, 70% lower with an uncertainty range of 64-75%). These results rely on the assumption that charging stations purchase grid electricity instead of using specific behind-the-meter sources or entering into specific power purchase agreements.

### Social cost – combined private and external cost

Figure 4 displays the total social costs per VMT, or combined private and external costs, of long-haul BE-HDV and diesel HDVs in (A) 2025 and (B) 2035 under Low RE. Figures S22–S36 display these results under both renewable energy cost scenarios, all cathode chemistries, and different discount rates. Costs are assigned to three categories: "External - GWP" representing the external costs from GHG emissions, "External - Human Health" representing the human health damages from local air pollutants, and "Private" representing direct monetary costs. Additionally, the external costs assigned "GWP Truck" are grouped in "General Ops"; "GWP Elec. Gen. and Loss" as well as "GWP Elec. Upstream" costs are grouped in "Electricity"; and "GWP Battery" costs are grouped in "Battery". "Payload" represents the private costs

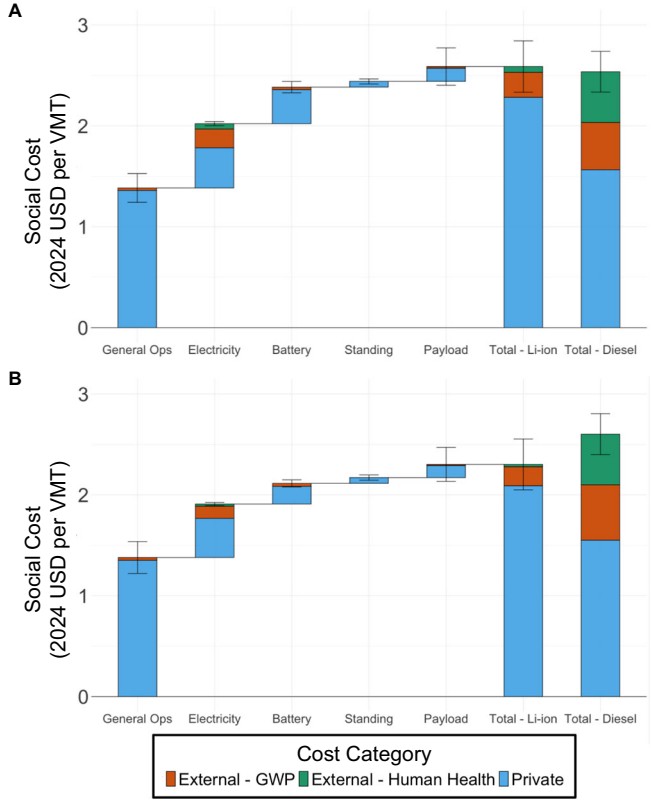

**Fig. 4 | Social costs of long-haul BE-HDVs and diesel HDVs in varying years and renewable energy cost scenarios. A** Visualized social costs in 2025 under Low Renewable Energy Cost. The difference in the social cost of Li-ion BE-HDVs and diesel HDVs is $0.05/VMT. **B** Visualized social costs in 2035 under Low Renewable Energy Cost. The difference in the social cost of Li-ion BE-HDVs and diesel HDVs is -$0.31/VMT. Uncertainty bars represent two standard deviations.

from offset cargo due to the added battery weight and the external costs induced from the additional trip incurred to transport this off-set cargo.

The significance of external costs is notable for diesel trucks in 2025 and 2035; GWP damages account for approximately 20% of total social costs while human health damages contribute another 20%. Specifically, social costs from human health damages are approximately $0.50 per VMT, central to the range of non-forecasted estimates in literature[26]. Variance may arise from the VSL and SCC values which introduce additional uncertainty beyond what is shown in the uncertainty bars for this study. A more in-depth exploration of this uncertainty warrants its own study since variations in the SCC and VSL do impact social costs. A more in-depth comparison of our values to literature is provided in the Methods. The social costs for BE-HDVs, conversely, are dominated by private costs with external costs comprising <12% of total social costs in 2025 and 2035. In 2025 under Low RE, the external cost advantages of BE-HDVs do not yet outweigh the private cost premium. The social cost of operating a BE-HDV is $0.05/VMT higher than diesel. Under High RE (Fig. S22), the difference in social cost increases to $0.11/VMT due to a decreased likelihood of renewable energy offsetting fossil fuel combustion as the marginal electricity source.

By 2035, the anticipated reduction in battery prices and increased penetration of renewable energy on the grid alters the conclusion, with BE-HDVs expected to achieve lower social costs than diesel. Under Low RE, the social benefit of a long-haul BE-HDV relative to diesel is $0.31/VMT, and under High RE (Fig. S22), the social benefit is $0.22/VMT. Although these results are uncertain and sensitive to changes in battery prices, the values suggest that BE-HDVs are currently within a small margin of achieving social benefit relative to diesel HDVs and even smaller-than-anticipated improvements in costs could be sufficient to tip the balance towards BE-HDVs within several years.

## Discussion

This study offers definitive insights into whether there is a substantial social benefit in 2025 and 2035 from electrifying long-haul diesel HDVs with BE-HDVs. The modeled private costs of long-haul BE-HDVs are substantially higher than diesel HDVs in 2025 and 2035; $0.72/VMT and $0.53/VMT higher respectively. However, these costs are particularly uncertain due to the unpredictability of future charging infrastructure costs and fuel prices. Our results show that the private costs of BE-HDVs as of 2025 are expected to remain higher than diesel HDVs at any electricity price greater than $0.01/kWh given constant diesel prices. However, by 2035, it is possible for the private costs of BE-HDVs to dip below those of diesel HDVs if electricity prices at the charging station are below $0.14/kWh and battery prices decrease based on a 17.3% learning rate. Additionally, it may be possible for the private cost of diesel HDVs to exceed BE-HDVs at the modeled current electricity price if diesel prices exceed $8.00/gal. While our sensitivity analysis on charging and diesel prices aims to quantify the effects of uncertain future charging and fuel prices, it may not fully capture possible outcomes considering the current lack of infrastructure for this EV use case, regional and temporal variability in electricity prices, and trends in long-term electricity portfolios. Figure S14 and S15 further visualize the impacts of variable charging prices on our results. The recovery value of the degraded Li-ion battery pack may also impact BE-HDV private costs. We conservatively assume the full depreciation of batteries, resulting in relatively high private costs for BE-HDVs compared to literature[14,15,27–32] but more accurately reflecting current battery circularity practices in the US[33–36]. In general, the pricing of Li-ion batteries into the future is uncertain and may impact the private costs of BE-HDVs due to their reliance on critical materials, unstable international market dynamics, and their potential for technological improvements.

In contrast to private costs, long-haul BE-HDVs have reduced external costs relative to diesel HDVs across all years and scenarios examined. While geography may influence the magnitude of the external costs for diesel and BE-HDVs, at a national scale the external costs of BE-HDVs remain substantially below diesel. Figures S37–S40 illustrate the differences in external costs that occur when operating BE-HDVs in different regions in the US, displayed in Fig. S41. Across all national scenarios, BE-HDVs see external benefits between $0.62/VMT and $0.84/VMT relative to the use of diesel HDVs. This benefit, however, is dependent on the methodologies and values used to estimate external costs. Given that this study uses higher SCCs and modern models of human health burden, we find our estimates of the external costs for diesel HDVs to be higher relative to past literature[26,30,37–39]. The SCC in particular is highly uncertain and official government estimates can vary widely. Additionally, the ISRM used to translate emissions to air pollution impacts in this study is designed for computational efficiency but it makes several simplifying assumptions, as discussed in greater detail in McNeil et al.[7] and Tessum et al.[24]. The ISRM also relies on background concentration and demographic data that is not updated to the current year or future years. Updating this data may improve results but is a tremendous undertaking for a study not primarily focused on background concentrations and demographics; thus it is not standard practice throughout literature[7,21,40,41]. Development of standard scenarios for future demographic changes and background air pollutant concentrations would be of great value to the research community. Beyond potential ISRM updates, a more thorough exploration of uncertainty regarding the VSL and the magnitude of the mortality effects of ambient $PM_{2.5}$ exposure would improve the quality of similar studies in the future.

Several modeling choices on the behaviors of BE-HDVs and the supporting electrical infrastructure introduce additional uncertainties that may impact results. While drive cycles are an acceptable proxy for vehicle behavior, they are not fully representative of real-world driving decisions and may mischaracterize the energy efficiency of HDVs, particularly for the impacts of regenerative braking which are highly dependent on driver behavior and road conditions. We provide Fig. S46 to explore how our results vary under a simplified drive cycle with no regenerative braking savings. Our study also employs a simple battery performance and degradation model that estimates cycling and calendar aging, but omits factors like regional temperature, driver behavior, road conditions, local traffic flow, etc. A physio-chemical model of battery performance that captures the impacts of these factors may improve results but comes at the cost of increased computational demand. Additionally, due to our use of a short-run marginal generation model, our per-VMT emissions and air pollution results can be used for comparatively small increases in grid loads but may not be representative for larger loads requiring major shifts in electricity generation. We cannot say with certainty the maximum scale beyond which our per-VMT results would no longer be representative of realistic grid responses and emissions.

In order to further explore the potential costs and benefits of BE-HDVs, future work should examine mechanisms for BE-HDVs adoption and explore the impacts associated with higher rates of adoption, as the results in these studies represent the operation of a marginal truck. Future research should also consider examining which communities receive the greatest benefits or costs associated with BE-HDV adoption to better understand potential equity and justice implications. Although this study focuses on the US, the methods can be extended to other countries and regions, provided accessible methodologies for connecting charging loads to power plant emissions and modeling of air pollution related human health impacts are available. Additionally, fully capturing the impacts of Li-ion battery production requires models capable of capturing these global supply chains[42]. Different countries have distinct electricity mixes, population distributions, background concentrations of emissions, and electricity rate

structures that may substantially alter the balance of costs and benefits for BE-HDVs. Researchers should also continue to compare the private and external costs of alternative zero emission HDVs performing long haul freight relative to diesel HDVs. Hydrogen fuel cell HDVs and BE-HDVs with battery swapping capabilities are of particular interest due to their declining technology costs and their potential to avoid costs associated with long recharge times. While literature is beginning to explore the impacts of these technologies[28,38,43–47], their externalities remain relatively unexplored.

Despite the high external benefits and greater private costs of BE-HDVs relative to diesel HDVs, current federal tax credits only provide a $40,000 incentive to purchase BE-HDVs, equivalent to $0.09/VMT[48]. This incentive, assuming its continued availability, still falls short of what is needed to offset the elevated private costs BE-HDVs and is substantially less than the monetized net external benefits of using diesel HDVs instead of BE-HDVs. While our results indicate that the combined private and external benefit of BE-HDVs relative to diesel HDVs is negative in 2025, meaning the elevated private costs are not fully offset by the external benefits, the net social benefit of BE-HDVs by 2035 is projected to range between $0.22/VMT to $0.31/VMT depending on the renewable energy cost scenario. Based on the limited rate at which truck fleets turn over[42], incentives and/or mandates are required years in advance of 2035 to achieve appreciable uptake and social benefits by then.

Additionally, if one assumes the sole purpose of the federal incentive is to avoid $CO_{2eq}$ emissions, we find that the $40,000 tax credit values the cost of $CO_{2eq}$ between $80.36/tonne and $67.62/tonne in 2025 depending on the renewable energy cost scenario, and between $62.05/tonne and $49.84/tonne in 2035, well below the EPA's estimate of the SCC[22]. By reversing this exercise and using the EPA's 2025 estimate of the SCC with a 2% discount rate, we find that the incentive for purchasing a BE-HDV, if based solely on the SCC, is $106,000 to $125,000 or $0.25/VMT to $0.29/VMT depending on the renewable energy cost scenario. Using the EPA's 2035 estimate of the SCC, the incentive is $160,000 to $199,000 or $0.36/VMT to $0.45/VMT depending on the renewable energy cost scenario. However, if we include these idealized incentives in our NPV analysis, the private cost of BE-HDVs still remain higher than diesel HDVs in 2025 and 2035. Thus, to promote the adoption of BE-HDVs and realize their expected social benefits, incentives need to go beyond the value of GHG mitigation alone by incorporating the value of reductions in human health damages.

## Methods
### Study boundary and modeling decisions
This study provides a comprehensive comparison of the private and external costs of battery-electric heavy-duty vehicles (BE-HDVs) performing long-haul freight relative to diesel heavy-duty vehicles (HDVs). We achieve this by performing both a technoeconomic analysis and a life-cycle assessment of BE-HDVs and diesel HDVs. Included in these analyses are the impacts from raw material production/consumption, electricity use, thermal fuel use, greenhouse gas emissions, and criteria air pollutant emissions across several main life-cycle phases of BE-HDVs and diesel HDVs: (1) Li-ion battery production and assembly, (2) HDV production and assembly, (3) regional electricity generation, and (4) HDV long-haul freight. We exclude the impacts associated with downstream lifecycle phases including regional/local freight, drayage freight, and end-of-life. We assume transportation impacts are negligible and that differences between non-tailpipe emissions for diesel and BE-HDVs are negligible due to the high uncertainty regarding the opposing impacts of regenerative braking and increased vehicle weight on brake and tire wear (BTW) emissions. While we acknowledge that BTW is a non-negligible contributor to $PM_{2.5}$ emissions[49,50], we find that there is high uncertainty and little consensus in current literature on how to appropriately model the BTW emissions specifically for BE-

HDVs given these contributing factors[51–53]. This topic is worthy of further study, and requires more emissions measurements to determine the most realistic assumptions.

When modeling Li-ion battery production and assembly, the methods used in Porzio et al.[54] and the material quantities outlined in Porzio and Scown[36] are employed to describe the raw material consumption, electricity and energy usage, and emissions associated with Li-ion batteries used in BE-HDVs. The NMC811 cathode chemistry is used in the results of the main text, but the impacts associated with NCA and LFP cathode chemistries are modeled and presented in the Supplemental Information. Additionally, Tables S4 and S5 are used to determine the raw material consumption, electricity and energy usage, and emissions associated with HDV production and assembly. Human health impacts associated with manufacturing and material extraction are omitted in this procedure due to the high levels of uncertainty regarding the location and fuel mixes of these activities. Modeling decisions, impacts, and costs associated with the HDV long-haul freight use phase and regional electricity generation are explored in the following sections.

### Literature review and comparison
The private and external costs of BE-HDVs and diesel HDVs modeled in this study are compared against costs throughout literature in Fig. S1, with all reported costs converted to 2024 USD per VMT. In Fig. S1A, the modeled diesel HDV private costs in this study ($1.57 per VMT in 2025 and 2035) are relatively central to the range of private costs across literature, matching the mean literature private cost of $1.57 per VMT, although variations in discount rates occur across the literature[14,15,27–32,55]. We expect some spread in literature values given the variety of methodologies, boundaries, and assumptions used to determine private costs of diesel HDVs. The literature sampled includes studies focused on both European and US regions, varying assumptions regarding ownership length and operational time in long-haul freight, as well as fuel prices that reflect when the studies were conducted. Hunter et al.[14] employs methods most similar to those used in this study and expects that the private costs of long-haul diesel HDVs in 2025 would range between $1.39 and $1.58 per VMT; our modeled private cost falls within this range. Additionally, we excluded the 2050 private costs from comparison, as these all represent "ultimate" estimates from Hunter et al.[14]

The modeled private costs of BE-HDVs in this study ($2.28 and $2.09 per VMT in 2025 and 2035 respectively) are visually in the upper range of private costs for BE-HDVs in literature, falling within one standard deviation above the mean value of $1.76 per VMT in 2025[14,15,27–32,55]. This difference is primarily driven by assumptions regarding the residual value of battery packs in BE-HDVS. We assume no residual value on a used battery pack since the viability of second-life battery application and battery recycling are still under-developed in the US[33–36], while some sources in literature assume residual values as high as 49%[27].

In Fig. S1B, the modeled external costs of diesel HDVs in this study ($0.97 and $1.05 per VMT in 2025 and 2035 respectively) fall into the upper range of estimates across literature[26,30,37–39]. Our elevated external cost for diesel can primarily be attributed to three reasons: (1) The SCC used in studies has increased substantially overtime[22,56–58], resulting in greater climate damages attributable to the same emissions; (2) improved modeling and understanding of local air pollution and population dynamics[24,25,59–61] contributing to greater human health impacts; and (3) varying geography and boundaries on external costs in literature, with some studies excluding climate damages[26] and others including damages to infrastructure, traffic congestion, noise, accidents, non-particulate pollutants, etc[30,37–39]. If we ignore the climate damages, which are highly variable depending on the SCC used, and compare the human health damages of the diesel HDVs in this study ($0.50 per VMT) to those in Cohon et al.[26] whose boundaries are

most similar to this study's, we find that our results are central to their non-forecasted estimates (between -$0.008 and $0.94 per VMT in 2005) despite Cohon et al. using higher emission factors representative of older pollution control technologies in HDVs. We find that a decreased population size at the time of analysis and their use of a simpler air pollution model counteract their elevated damages, resulting in similar net damages.

While there is an increasing body of literature on the modeling of external costs of BE-HDVs[7,20,21,62], very few studies normalize their results to functional units relevant for comparison. Regardless, the external costs of BE-HDV modeled in this study visually align well with the little values available in literature[38]. For future studies, we recommend that results for external costs are presented in dollars per VMT to allow for integration with private cost or total cost of ownership analyses. For comparison, we normalize the human health damages for BE-HDV in this study by the electricity demand induced on electrical infrastructure, visualized in Table S1. Relative to the monetized human health damages of electricity from Cohon et al.[26] (Table S2), the 2025 damages from electricity generation in this study are central to the Cohon et al.[26] damages from coal, but decrease below the mean by 2035, indicating less reliance on more polluting sources of electricity. This comparison is consistent with our use of a marginal short run marginal generator, as per Discussion S1.

## Truck power model and design parameters

A model of battery electric truck power demand and performance is required to model energy consumption associated with trucking behavior. Equations 1 through 6 make up the standard model for the instantaneous power demand of a vehicle used in this study[63].

$$P_{Tot} = \left( \frac{P_{AR} + P_F + P_G}{\eta_{BW}} + P_I * \left( \frac{1}{\eta_{BW}} - \eta_{BW} * \eta_{BR} \right) + P_{AC} \right) * (1 - R_{Br})$$

(1)

$$P_{AR} = 0.5 * \rho * C_D * A * v^3$$

(2)

$$P_F = C_{RR} * m * g * v$$

(3)

$$P_G = m * g * v * Z$$

(4)

$$P_I = 0.5 * m * v * a$$

(5)

$$m = \min(m_B + m_V + m_P, GVWR)$$

(6)

A time resolution of a minute is used in Eqs. 1 through 6. All variables represent the average value over this resolution. In these equations $P_{Tot}$ represents the total power demand, $P_{AR}$ represents the power contribution due to air resistance, $P_F$ represent the power contribution due to friction, $P_G$ represents the power contribution due to gravity, $P_I$ represents the power contribution due to inertia, and $P_{Ac}$ represents the power contribution from accessory loads. Since the impacts from regenerative braking typically occur at a time resolution less than a minute, we instead model the energy recovered as reduction to power demand. The value of this reduction ($R_{Br}$) is determined via review of reported regenerative braking performance of electric class 8 trucks commercially available today. $\eta_{BW}$ and $\eta_{BR}$ represent the battery-to-wheels efficiency and brake efficiency respectively. $\rho$ represents air density, $C_D$ represents the vehicle's drag coefficient, A represents the frontal area of the truck, $v$ represents the velocity of the truck, $C_{rr}$ represents the coefficient of rolling resistance between the truck tires and the road, $g$ represent the gravitational coefficient, $Z$ represents the current grade of the road, and $a$ represents the

acceleration. Additionally, $m$ represents the total weight of the vehicle consisting of the weight of battery ($m_B$), the weight of the tractor and trailer excluding the battery weight ($m_V$), and the weight of the payload ($m_P$). The maximum total weight of the vehicle is set at the maximum federal gross vehicle weight rating (GVWR) of 82,000 lbs for Li-ion class 8 trucks and 80,000 for diesel class 8 trucks[14,15].

Table S3 shows the parameter values and battery size, as well as the resulting range of the vehicles included in this study. Uncertainties for all these parameters are presented in Table S6. Battery weight varies by the battery chemistry chosen and is determined by multiplying the chemistry-specific pack specific energy by the battery size. The pack specific energies for each chemistry are provided in Table S7[64].

The design parameters and power demand model of diesel class 8 trucks are similar to Li-ion BE-HDV with $\eta_{BW}$ being replaced by the product of the diesel engine efficiency ($\eta_E$) and the transmission-to-wheels efficiency ($\eta_{TW}$). The parameters for the diesel HDV design scenario are also displayed in Table S3[20,63,65]. To simplify modeling and have more focused results, we model all HDVs as class 8 trucks, which make up over 80% of new HDV truck sales as of 2021[3].

Due to computational limitations, road grade ($Z$) is simplified to the average road grade between two nodes of the trucking corridor network detailed in the following section. Future models with greater computational resources and increased time resolutions may see improved accuracy in model results by using instantaneous road grades.

While temperature (particularly cold environments) has negative impacts on short-term battery performance[66,67], industry interviews[16,68] highlight the efficacy of thermal management strategies and heat generation of Li-ion batteries in reducing these impacts, especially for long-term operation like long-haul freight transport. Additionally, the minor impact of HVAC loads relative to energy demand from locomotion in heavy-duty freight, the infrequency of extreme temperature events in most regions, and the high computational resources required for an accurate thermodynamic model led us to exclude the impacts of temperature in our behavioral model.

## Trip generation and behavioral model

We constructed a model to describe the trips performed by a truck and how it would complete them over a year of operating in long-haul freight. To determine the routing of a HDV, we referenced the 2017 Commodity Flow Survey (CFS)[69] and only observed trips that were over 250 miles and performed by class 8 trucks. The CFS provides origin-destination (OD) pairs between all major metropolitan areas and describes characteristics for typical trips along these OD pairs. These characteristics include payload weight (Fig. S42), whether the trailer was refrigerated, and a statistical-weight parameter that describes how frequently trips had these characteristics and how frequently trips occurred between OD pairs. These statistical weight parameters were used to construct discrete probability density functions (PDFs) describing the likelihood of going to specific destinations from different origins, the likelihood of common payload weights, and the likelihood a trailer was refrigerated.

We assigned the major metropolitan areas from the CFS as nodes in a network of all major US trucking corridors at a 1 km scale and set the corridors as links[70]. Additionally, nodes where charging could occur were placed at every major metropolitan area and every 250 miles along corridors. The length of corridors and road grade were stored in these links. The constructed PDFs were then used to determine an OD pair, a payload weight, and whether the trailer was refrigerated. A shortest path algorithm was used to determine the specific routing a truck would follow within the corridor network. This procedure was repeated, using the previous destination as the origin for the next trip. This was performed for 100 trucks each operating for a year, resulting in the truck flow outlined in Fig. S43. If the chosen

payload weight for a trip causes the total vehicle weight to exceed the federal GVWR limit, we assume that the excess weight is removed from payload and must be shipped by taking up a portion of another vehicle's cargo space. These excess tonne-miles are considered as a "Payload Cost" when calculating ownership costs.

A behavioral model of Li-ion BE-HDVs and diesel HDVs is used to determine dispatch times, vehicle speeds, when and how charging occurs, and when to rest while completing the routes determined from the trip generation model for a year of operation. The outputs of this model included the VMTs and power demand along every kilometer driven in the network, the hourly electricity power demand at every node where charging occurs, time spent driving, time spent simultaneously resting and charging, and time spent solely charging over the course of a year. Initial trip dispatching from a location was decided via a discrete probability density function presented in Fig. S44[71]. We assume truck velocity and acceleration followed the California Air Resource Board Heavy Heavy-Duty Diesel Truck Cruise Segment drive cycle when driving[72]. This drive cycle is visualized in Fig. S45. Results with a simplified drive cycle (constant 65 mph, or approximately 105 km/h) are presented in Fig. S46. Additionally, our model employed logic to ensure that the state-of-charge of a Li-ion battery on a class 8 truck remained between zero and one, and that the labor regulations for truck drivers in the US were followed[73]; specifically, limits on consecutive hours spent driving and requirements for rest. We assume there is no preference on the time of day when charging occurred and that the BE-HDV would choose to drive over charging unless it could not complete a segment with its remaining state-of-charge. Charging speeds as a function of state-of-charge and infrastructure type are presented in Fig. S47[68,74,75]. An example of HDV behavior throughout a day is visualized in Fig. S48. Similar logic is used to determine diesel HDV behavior, with a minor time penalty of 15 minutes when refueling occurs.

## Grid emissions modeling and external costs

The hourly electricity power demand at charging nodes is used to determine the use phase contributions to GWP and human health burden from Li-ion BE-HDV operation. We employ the short-run marginal generator electricity grid model outlined in McNeil et al.[7,21] to determine marginal generator type and location responding to the hourly electricity demand at each node. This model considers power flow between regions at an hourly resolution and how the grid evolves over the next several decades due to renewable energy sources taking over much of the marginal generation in many regions. Specifically, this grid model uses NREL's Cambium datasets[67,71] to estimate how hourly marginal generation is projected to change by region into the future. Justification and implications of using a marginal short-run generator model are provided in Discussion S1.

Charging loads are assigned to a balancing area by geography which are then assigned to a transmission connected region (T-region) made up of several balancing areas, although the specific balancing areas that make up any given T-region change hour-by-hour depending on where power is predicted to flow[76]. The charging loads are then allocated to all the power plants in that T-region classified as the marginal generator type. We then assume that the charging load is distributed between each generator, such that the induced load is proportional to their current generation[77]. While the approach is imperfect, it provides a decent estimate for which generators are responding to a theoretical load. For greater detail, see McNeil et al.[21].

The emitted GHGs and criteria air pollutants at the responding marginal generators are then estimated using generator-specific emission factors from the Grid Optimized Operation Dispatch Model developed by Jenn et al.[77]. GHG emissions are then converted to tonnes of $CO_2$-equivalents ($CO_{2eq}$) to determine contributions to GWP. Emissions of primary particulate matter ($PM_{2.5}$) and relevant precursors for formation of secondary $PM_{2.5}$ are used to determine resulting human

health damages using the InMAP source–receptor matrix (ISRM)[24,25,60,78]. Two scenarios of renewable energy adoption are used to determine responding marginal generators: one where renewable energy costs are high (High RE) and one where renewable energy costs are low (Low RE). These scenarios are outlined in NREL's Standard Scenarios[79,80].

For diesel HDV operation, we use the modeled VMT and power demand along each kilometer segment of all trucking corridors to determine contributions to GWP and human health burden following procedure outlined in McNeil et al.[21]. Emission factors from GREET[81] and Preble et al.[82] are used to determine GHG and $PM_{2.5}$ emissions at each kilometer segment. GHG emissions are then converted to tonnes of $CO_2$-equivalents ($CO_{2eq}$) to determine contributions to GWP. Two sets of emission factors are used: one representing the current performance of pollution control technologies (diesel particulate filter and selective catalytic reduction), and one representing the performance of potentially improved pollution control technologies (although the improved performance is notably marginal due to limitations of pollution control technologies)[20,81–83]. Table S8 contains the specific emission factors used. Emissions of primary $PM_{2.5}$ and relevant precursors are used to determine resulting human health damages using the ISRM[24,25,60,78]. For external costs in 2024, we assume emission factors match those of model year 2010 through 2018 diesel class 8 trucks due to the physical and chemical constraints of the pollution diesel particulate filters and selective catalytic reduction that would limit substantial changes in EFs by year absent any disruptive change in control technologies. Following this logic, we assume only incremental improvements to 2024 emission factors when extrapolating to emissions and external costs in 2035, as per Table S8. These assumptions align with Tong et al.[20] and avoid reliance on arbitrary assumptions about emission standards, policies, and their impacts on real-world vehicle performance.

Annual mortality from $PM_{2.5}$ emissions is modeled using the concentration-response functions from Krewski et al.[60] and Lepeule et al.[61], provided in Eqs. 7 and 8 respectively, where $Deaths_K$ and $Deaths_L$ represent the number of deaths using the Krewski and Lepeule equations respectively. Additionally, $PM_{2.5}$ represents emissions in kg, $Population$ represents the total population per grid cell, $Mortality Rate$ represents the year 2005 baseline overall population mortality rate in deaths per year per 100,000 people and mortality rate increases by 6% and 14% for every 10 μg/m³ $PM_{2.5}$ for the Krewski et al.[60] and Lepeule et al.[61] equations respectively. Tessum et al.[24], Goodkind et al.[25] and Thind et al.[78] provide greater detail on how these concentration-response functions are integrated into the ISRM. When modeling human health damages, a $11.51 million value of a statistical life (VSL)[84] is applied to the mean deaths from the Krewski et al.[60] and Lepeule et al.[61] equations. Note that this VSL is used for results in 2025 and 2035 following procedures by Goforth and Nock[40] and Jackson et al.[41] to limit the uncertainty of future mortality valuations.

$$\text{Deaths}_K = \left( e^{\left( \frac{\log(1.06)}{10} * PM_{2.5} \right)} - 1 \right) * \frac{\text{Population} * \text{Mortality Rate}}{100,000} \quad (7)$$

$$\text{Deaths}_L = \left( e^{\left( \frac{\log(1.14)}{10} * PM_{2.5} \right)} - 1 \right) * \frac{\text{Population} * \text{Mortality Rate}}{100,000} \quad (8)$$

The above relationship between $PM_{2.5}$ and mortality may be a substantial source of uncaptured uncertainty, given regional and temporal variations in the vulnerability of local populations[85,86] as well as potential methodological errors tied to dose-response curve development[87]. These drivers of epistemic and aleatory uncertainty are estimated to contribute around +/−3% change in deaths for a given $PM_{2.5}$ concentration[85,86]. Additional sources of uncertainty are associated with the use of InMAP and are further discussed in the "Limitations" section. Furthermore, there is uncertainty surrounding the

VSL due to imperfect measures of risk and value used to estimate it, along with the inherent sampling bias present in complex statistical analyses[88,89]. A recent literature review demonstrated that the estimated VSL may vary by as much as +/−50% across different studies[89]. Further, estimation of the social cost of carbon also relies on complex climate models and damage functions for the many modes of climate impacts that have multiple sources of uncertainty. Recent estimates show uncertainty has a substantial impact on the SCC estimate depending on the discount rate used during calculation[22,58].

### Private costs
Private costs are broken into five major categories: General Ops, Fuel/Electricity, Battery, Standing, and Payload. We calculate the private cost after modeling truck behavior for a year and assuming four years spent performing long-haul freight. The annual VMT, age, and other statistics are used to model the categories of ownerships. Discount rates of 2%, 3%, 5% and 7% are used for determining costs into the future.

General Ops costs consist of vehicle depreciation, insurance, taxes, additional fees, maintenance, and driving labor. Vehicle depreciation is modeled as a percentage of the initial manufacturer suggested retail price of a vehicle. This percentage is determined by a function of the VMT and age of a vehicle and is assigned to occur at the end of the long-haul freight use phase[14,15]. The cost of the battery for Li-ion BE-HDVs is excluded from this calculation given that no substantial second-life battery market currently exists[90]. Insurance is modeled as an annual cost per annual VMT[15]. Taxes consist of an upfront component due to the federal excise tax and a fixed annual component due to the Heavy Vehicle Use Tax[91,92]. Fees consist of a variable annual component dependent on vehicle weight representing registration fees and a variable annual component dependent on VMT due to tolls and miscellaneous permits[15]. Maintenance consists of a variable annual component determined by a function of vehicle miles traveled and vehicle age. Additionally, maintenance costs are higher for diesel HDVs than BE-HDVs due to the increased complexity of their powertrain[14,15]. Driving labor is modeled as an annual cost per annual VMT. Table S9 describes the calculation and uncertainties of the values used to calculate general ops costs.

Fuel/Electricity costs for Li-ion BE-HDVs consist of two components: one representing the cost of the energy provided, and one representing the cost of utilizing provided charging infrastructure. The cost of the energy provided is set at $0.17/kWh to agree with US EIA's[18] estimate of electricity prices for transportation. The cost of utilizing charging infrastructure is set at $0.09/kWh as estimated by Burnham et al.[17] and primarily considers equipment costs, installation costs, infrastructure lifetime, and infrastructure provider margins. For diesel HDVs, the cost of fuel consists of the national average diesel prices as forecasted by the US EIA[18] at $3.75/gal.

Battery costs are only present for Li-ion BE-HDVs and assigned as an upfront cost. Current battery pack prices for each cathode chemistry are used when modeling 2024 costs of ownership. A learning rate of 17.3% and a forecast of Li-ion battery demand as used to model 2035 costs. Figure S49 and Tables S10 and S11 visualize the modeled future battery prices, battery demand, and uncertainty. The procedure used in Porzio et al.[54] is employed to size truck batteries such that they can maintain their rated power over the entire time spent operating in long-haul freight even as they degrade due to cycling and calendar aging. Additionally, we assume the batteries have no residual value at the end of the long-haul freight use phase given the current development of US battery circularity[33–36].

Standing costs for Li-ion BE-HDVs represents the additional cost of labor accrued while the truck is charging. This is modeled by multiplying the annual time spent charging (excluding time simultaneously used to rest for the driver) by the hourly cost of labor and supporting operations incurred from this additional time[15].

Payload costs for Li-ion BE-HDVs represent the cost of hauling the excess tonne-miles due to the offset cargo from the additional battery weight relative to diesel HDVs. The average excess tonne-miles is normalized by the payload capacity of an additional Li-ion BE-HDVs and multiplied by the total cost of ownership excluding payload costs[14,15]. All external costs are also scaled by this cost factor, thus accounting for the system costs induced by an additional BE-HDV operation. Given the scope of our study, we acknowledge that this approach may omit additional external costs associated with the need for a greater number of overall vehicles (i.e., increased congestion, greater BTW emissions, etc.), and suggest that future works further investigate these topics.

### Reporting summary
Further information on research design is available in the Nature Portfolio Reporting Summary linked to this article.

## Data availability
The processed Figs. 1–4 data is available at https://doi.org/10.6084/m9.figshare.28673372. The freight truck counts data are available from the 2017 Commodity Flow Survey Datasets at https://www.census.gov/data/datasets/2017/econ/cfs/historical-datasets.html (ref. 60). The electricity grid emissions calculations data are available from the National Renewable Energy Laboratory's Cambium tool at https://www.nrel.gov/analysis/cambium.html (refs. 67,71). The Source-Receptor Matrices used for health impacts estimation data are available at https://inmap.run/. The representative drive cycle data are available from the National Renewable Energy Laboratory's DriveCAT: Drive Cycle Analysis Tool at https://www.nrel.gov/transportation/drive-cycle-tool (ref. 63). Geographic data on contiguous United States boundaries and truck corridors is available from the United States Census Bureau at https://www.census.gov/geographies/mapping-files/time-series/geo/carto-boundary-file.html (ref. 93) and the United States Bureau of Transportation Statistics at https://data-usdot.opendata.arcgis.com (ref. 70) respectively.

## Code availability
The truck behavioral model, LCA, and TEA code are available via Code Ocean at https://doi.org/10.24433/CO.4346328.v1. Grid model code is available via Dryad at https://doi.org/10.5061/dryad.2280gb63j.

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

## Acknowledgements

This work was supported by the Energy and Biosciences Institute (EBI) through the EBI-Shell program (J.P., W.M., M.A., C.D.S.).

## Author contributions

J.P., W.M., M.A., and C.D.S. designed the study. J.P. constructed the model and performed primary analyses. W.M. modeled generator

responses and human health impacts. J.P drafted the paper. J.P., W.M., F.T., S.M., M.A., and C.D.S. edited the paper draft.

## Competing interests

The authors declare no competing interests.
