## [Transparent Peer Review file · Nature Communications]

Electrifying Long-Haul Freight Trucks Reduces Societal Costs in the United States

Corresponding Author: Dr Corinne Scown

Version 0:

Reviewer comments:

Reviewer #1

(Remarks to the Author)

This paper provides a comprehensive assessment of the societal cost of two different long haul freight vehicle powertrains (battery electric trucks and conventional diesel trucks) in the United States. It combines the private total cost of ownership with monetized damages of global warming and human health impacts. The results show that while battery electric trucks have higher private costs, both now and projected in 2035, they will have a lower societal cost in 2035 due to the higher global warming and human health impacts. It is also notable that both of these impacts must be considered to demonstrate the lower societal cost (if only global warming or only health impacts were considered, the diesel vehicle would have the lower cost). A few specific questions are below:

1. Since you model trucking corridors across the country, do you account for regional factors that would impact vehicle performance (e.g., grade, temperature)? Do you account for any regional cost differences (diesel prices, electricity prices)?
2. Line 187 - "Human health impacts included in this study are limited to long-haul HDV tailpipe emissions and electricity generator stack emissions due to the limited availability and high uncertainty of modeling human health damages from battery manufacturing and material extraction, which typically occurs outside the US." I understand why you have not included human health impacts from manufacturing and material extraction, but it may be beneficial to include human health impacts from vehicle operation beyond tailpipe emissions. Brake and tire wear can represent a significant portion of vehicle PM2.5 emissions.
3. Some clarification on the payload cost would be helpful. Line 594 – "Payload costs for Li-ion BE-HDVs represent the cost of performing the excess tonne-miles due to the offset cargo from the additional battery weight relative to diesel HDVs. The average excess tonne-miles is normalized by the payload capacity of an additional Li-ion BE-HDVs and multiplied by the total cost of ownership excluding payload costs." Is the assumption that excess weight can be shifted to a different vehicle making the same trip? And therefore the total number of vehicles in the diesel and electric scenarios are the same? It may be worth exploring whether a shift to electric trucks might necessitate an increase in the total number of vehicles (and this could be due to driver hour limitations and range limitations as well as cargo weight limitations) and how such an increase would change the results.
4. Based on figures S36 and S38, I'm somewhat concerned that the difference in fuel economy between the electric and diesel vehicle may be slightly overstated. The figures suggest that the long-haul truck is stopping multiple times every 30 minutes, and never surpassing 60 mph. In such a scenario, the electric truck would have a tangible benefit from regenerative braking compared to the diesel vehicle. However, the difference in fuel economy would be lower were the vehicles operating at a more steady rate. And I think the more steady operation may be a more realistic option.
5. The electricity price of \$0.26/kWh seem a bit low to me, just from comparing with the price currently offered at DC fast chargers. And building chargers capable of providing the energy required for long haul trucks will likely be even more expensive than constructing DC fast chargers. However, trucking companies may make agreements with electricity providers to achieve this price, so this might be fine.

6. It may be worth commenting on how these results can be applied to other settings (e.g., in different countries). I'm not suggesting any additional analysis of other locations, but you might discuss which inputs are likely to change or stay the same in other locations, and whether or not that would have a significant impact on the results.

Minor comments and suggestions

- Small typo in line 81 ("explores")
- The text and the figure captions skip from figure 2 to figure 4
- Citations 3 and 14 appear to be the same
- I'm assuming the 100-year GWP is used for the CO₂ equivalent of other GHGs. It might be good to state that explicitly somewhere in the methods section.
- In figure 5, it would be nice to have the years (2025 and 2035) for panels A and B shown directly on the figure.

Overall, this paper is thorough, and the methods are well documented in the text and in the supplemental information. It addresses an important topic and provides meaningful results for researchers, regulators, and fleet operators.

(Remarks on code availability)

Reviewer #2

(Remarks to the Author)

This paper aims to compare the private and social costs of electric heavy duty trucks with diesel heavy duty trucks. It estimates both private costs and air pollution externalities for the two vehicles. This is a very important and timely topic, especially as much of the research on the social costs and benefits of electric vehicles has focused on light duty electric vehicles, while heavy duty vehicles remain more challenging to electrify and less is known about their social benefits. However, I have important concerns about how the current paper estimates the external costs and therefore cannot recommend it for publication at this time.

Main Concerns

1. The paper provides little of the underlying data and assumptions that would be necessary to fully appreciate its results. It does not provide, for example, emission factors for the electric generating units; the concentration-response function used to link PM_{2.5} concentrations to mortality; the value per statistical life (VSL) used to monetize those impacts. It also does not provide the base year for any of these (e.g., what VSL applies to 2025 and 2035 and how was it VSL adjusted)

2. Emission factors for PM_{2.5} for diesel trucks seem extraordinarily high, and NO_x emission factors also seem high. NO_x and PM_{2.5} emissions are the pollutants that will cause almost 100% of the human health of the damages of the 2025 diesel trucks – the contribution from SO₂ would be negligible (In the case of the 2035 diesel truck it is possible that NH₃ emissions reported by the authors also contribute some).

On PM_{2.5}:

Table S9 provides 2.20 g PM_{2.5}/kg-fuel for 2025 and 0.30 g PM_{2.5}/kg-fuel in 2035. Using the fuel economies provided in Table S9 and assuming a density of 0.84 kg/L for diesel, this is equivalent to 1.30 g PM_{2.5}/mi in 2025 and 0.18 g PM_{2.5}/mi in 2035. The resulting value for 2025 is about 450 times higher (and even the 2035 value is about 60 times higher) than the 0.0028 g/mi PM_{2.5} (lifetime-weighted average) reported for GREET for a model year 2010 class 8 truck by Cai et al. (2015) for an earlier version of GREET [see Table 18, p. 42 in Cai et al. 2015]. Cai et al. also report 0.0032 g PM_{2.5}/mi in Tire and brake wear emissions. Even if the authors also meant include those (although the title of table S9 says "tailpipe"), the author's value for 2025 is still 200 times higher than total PM_{2.5} emissions in Cai et al. (the value for 2035 would be about 30 times higher)

On NO_x:

A similar comparison for NO_x suggests that the authors' values for 2025 are 30 times higher than the 0.2755 g/mile reported in Table 18 from Cai et al. (2015)

The values cited above are from an earlier version of GREET, as I was not able to access the GREET website at this time to provide with a citation for more a recent version of GREET. I acknowledge that especially NO_x emission rates may have been revised and substantially increased in more recent versions of the model.

Another comparison can be done with fleet-average values in 2025 reported for heavy duty vehicles (other than buses) by the Bureau of Transportation Statistics (BTS) (2025) based on EPA MOVES4. BTS (2025) reports:

PM_{2.5}: 0.054 g/mi (exhaust) (also 0.031 g/mi in brake wear and 0.003 g/mi in tire wear). NO_x: 2.942 g/mi.

The current paper's values are about 24 times higher for PM_{2.5} and 3 times higher for NO_x. I note, however, that this comparison with fleet average emission factors in 2025 is not adequate since fleet-wide emissions for PM_{2.5} and NO_x are much higher than for new model years as they are dominated by a small fraction of pre-2006 MY vehicles circulating which emit at very high rates. This is evidenced by the decreasing fleet-wide averages estimated by BTS through 2030, even in the absence of new major regulation. It seems difficult to understand how the authors estimate values higher than those fleet-wide averages.

3. Power plant damages also seem much higher than the literature.

For example, Dedoussi et al. (2020) estimates roughly 7,000 PM2.5-attributable deaths due to grid emissions in 2018. For a generation of about 4,200 TWh in that year (<https://www.eia.gov/todayinenergy/detail.php?id=38572>) and a VSL of roughly 12 million (US DHHS, 2021), this would be equivalent to 2 cents/kWh. Choma et al. (2024) also report a similar value of 2.2 cents/kWh for 2018 using InMAP – which further decreases over time. Furthermore, SO₂ emission rates per kWh dropped by 45% between 2018 and 2023, and NO_x emission rates by 33% (U.S. EPA, 2024a), so one would expect closer to 1 cent/kWh for 2025. The authors, on the other hand, report 5.8 cents/kWh for low RE and 6.6 cents/kWh for high RE (Table S1). Part of the difference could be explained by their choice of a marginal model, though it still seems difficult to reconcile damages per kWh that are an order of magnitude higher than suggested in the literature. Perhaps even more importantly, the electric truck would not consume the marginal electricity in 2025 but rather the marginal electricity at the time it is charged – over the next 15 years or so. This would likely decrease their health damages substantially if forecasted decreases in grid emission rates are realized.

4. The InMAP model used by the authors calculates damages using 2011 data – not just on emissions but also e.g. on the VSL (Goodkind et al., 2019) and I believe on demographics and mortality rates as well. The authors should explain how they applied 2011 InMAP values to 2025 and what adjustments, if any, were made.

5. The authors compare their results with the National Academies report by Cohon et al. (The authors' reference number 28) (National Research Council, 2010). However, the report by Cohon et al. uses 2005 emissions data (as well as a much simpler air pollution model) which renders the comparisons difficult to understand. For example, in Table 2-11 (p. 97) Cohon et al. mentions an average of 12 lb. SO₂ (a value of 10.1 is also given in p. 108 of Cohon et al) and 4.1 lb. NO_x per MWh for coal plants in 2005. These values are several times higher than what the US EPA (2024a) estimates for the average coal plant in 2023 (1.7 lb SO₂/MWh and 1.2 lb NO_x/MWh) – not to mention that the coal share of generation in 2023 is much lower than it was in 2005.

The difference is perhaps even larger for vehicles. Table D-9, Appendix D, p. 462 in Cohon et al. give 0.36 g PM_{2.5}/mi and 12.89 g NO_x/mi for a Heavy Duty Diesel Vehicle Class 8a and 0.36 g PM_{2.5}/mi and 15.10 g NO_x/mi for a Class 8b HDDV. These values are about 1-2 orders of magnitude higher than today (see values in point 3. Above).

In this case, while it seems difficult to fully appreciate any comparisons of the present study by Cohon et al., it is surprising that they yield similar externalities per mile despite drastic changes in emissions over the past 20 years.

6. I appreciate the authors' attempt to quantify the uncertainty about several components of the model; however, uncertainty about the externalities is not discussed. Uncertainty about several aspects included in the calculation of the external costs (e.g. the dose-response between PM_{2.5} and mortality, especially for diesel vehicles) may be one of the largest contributors to the uncertainty in their results. While it may be difficult to provide with numerical values for this uncertainty, this should be at least discussed. The uncertainty in the Social Cost of Carbon is also about an order of magnitude (Rennert et al., 2022) and does not seem to be incorporated by the authors.

Other concerns and comments

7. The authors should explain better the rationale for discounting different types of impacts with different discount rates. As one example, Lines 184-186 mention a 5% discount rate for health and 2% for the social cost of carbon. However, about half of the social cost of carbon is thought to be increased mortality from extreme heat (Rennert et al., 2022) so it is unclear why the two impacts merit different discount rates. The authors also do not explain whether they account for a cessation lag between changes in PM_{2.5} exposure and mortality effects (e.g., US EPA 2024b).

8. It is not clear why the authors use average of MYs 2010-2018 to characterize a MY 2024 diesel vehicle. It seems to me that a relevant comparison should be of a new EV relative to a new diesel.

9. The paper should mention some of the key limitations at the end

10. The authors should check the y axis labels in Figs 2B-2C – it seems they probably mean USD per gallon. And in Fig 4A-4B is ng/m³ of PM_{2.5} correct ? (or is it ug/m³)

References

- Bureau of Transportation Statistics. (2025). Estimated U.S. Average Vehicle Emissions Rates per Vehicle by Vehicle Type Using Gasoline, Diesel, and Electric <https://www.bts.gov/content/estimated-national-average-vehicle-emissions-rates-vehicle-vehicle-type-using-gasoline-and> (accessed 11 June 2025)
- Cai, H., Burnham, A., Wang, M., Hang, W., & Vyas, A. (2015). The GREET Model Expansion for Well-to-Wheels Analysis of Heavy-Duty Vehicles. <https://www.osti.gov/servlets/purl/1212730> (accessed 11 June 2025)
- Choma, E. F., Robinson, L. A., & Nadeau, K. C. (2024). Adopting electric school buses in the United States: Health and climate benefits. *Proceedings of the National Academy of Sciences*, 121(22), e2320338121. <https://doi.org/10.1073/pnas.2320338121>
- Dedoussi, I. C., Eastham, S. D., Monier, E., & Barrett, S. R. H. (2020). Premature mortality related to United States cross-state air pollution. *Nature*, 578(7794), 261-265. <https://doi.org/10.1038/s41586-020-1983-8>

Goodkind, A. L., Tessum, C. W., Coggins, J. S., Hill, J. D., & Marshall, J. D. (2019). Fine-scale damage estimates of particulate matter air pollution reveal opportunities for location-specific mitigation of emissions. *Proceedings of the National Academy of Sciences*, 116(18), 8775-8780. <https://doi.org/10.1073/pnas.1816102116>

National Research Council. (2010). *Hidden Costs of Energy: Unpriced Consequences of Energy Production and Use*. Washington, DC: The National Academies Press. <https://doi.org/10.17226/12794>

Rennert, K., Errickson, F., Prest, B. C., Rennels, L., Newell, R. G., Pizer, W., Kingdon, C., Wingenroth, J., Cooke, R., Parthum, B., Smith, D., Cromar, K., Diaz, D., Moore, F. C., Müller, U. K., Plevin, R. J., Raftery, A. E., Ševčíková, H., Sheets, H., . . . Anthoff, D. (2022). Comprehensive evidence implies a higher social cost of CO₂. *Nature*, 610(7933), 687-692. <https://doi.org/10.1038/s41586-022-05224-9>

U.S. Department of Health and Human Services (DHHS). (2021). Appendix D: Updating Value per Statistical Life (VSL) Estimates for Inflation and Changes in Real Income. <https://aspe.hhs.gov/sites/default/files/2021-07/hhs-guidelines-appendix-d-vsl-update.pdf>

U.S. Environmental Protection Agency (EPA). (2024a). Clean Air Power Sector Programs. <https://www.epa.gov/power-sector/progress-report-emissions-reductions> (accessed 11 June 2025)

U.S. Environmental Protection Agency (EPA). (2024b). Estimating PM_{2.5}- and Ozone-Attributable Health Benefits: 2024 Update. <https://www.epa.gov/system/files/documents/2024-06/estimating-pm2.5-and-ozone-attributable-health-benefits-tsd-2024.pdf>

(Remarks on code availability)

Version 1:

Reviewer comments:

Reviewer #1

(Remarks to the Author)

The authors have addressed each of my comments and have made important clarifications and significant improvements to their manuscript.

(Remarks on code availability)

Reviewer #2

(Remarks to the Author)

I appreciate the authors' efforts to address my questions, but I still have a few concerns about the manuscript

1. In their rebuttal, the authors confirm that their large externality values for electricity generation are a consequence of applying a marginal model. However, they do not disclose the emission rates involved. Without having an idea of the emission rates, it is not possible to understand the authors' results. Both users of their results to inform policies and other researchers building on their study aiming refining their model would need to have a notion of the emission rates involved to be able to compare and reconcile any possible differences in findings. While I understand that emission rates are drastically different for each EGU, a summary table of the resulting average emission rates per kWh (the average across all EGUs meeting the BE-HDV demand) would still be quite useful to the readers, and even the paper by Jenn et al. cited by the authors has published average emission rates under different scenarios.

2. The authors need to describe in a little bit more detail the "marginal" scenario assumed for electricity. The authors mention in line 630 that they simulate 100 trucks. There are more 3-4 million class 8 trucks alone in the US (e.g. <https://www.bts.gov/browse-statistical-products-and-data/surveys/vius/vehicle-stats-state-vehicle-type-and-gvwr>). If this entire truck fleet were to be electrified, they would increase total US electricity consumption by on the order of 10% (~2 kWh/miles x ~200 billion VMT/year[1] = ~4e11 kWh).
[1] <https://www.bts.gov/browse-statistical-products-and-data/freight-facts-and-figures/vehicle-miles-traveled-highway>

While electrifying 3-4 million trucks would not be realistic, especially in a short time frame, any realistic policy scenario would also involve more than 100 trucks. Therefore it raises the question as to whether the authors calculated the short run marginal to meet the additional demand of just 100 trucks – which probably would not be relevant to any realistic policy scenario (Especially as far as results for the future/2035, when it is not clear that meaningful adoption of electric HDVs would be met with existing electricity generating units/current short-run marginal).

2.1. The fact that all of the authors results are based on simulating just 100 trucks should be clarified earlier in the manuscript. Without that information it is difficult, for example, to make sense of the ~ 0.01 ng/m³ change in PM_{2.5} in figures 4A/4B (which is only clear once one realizes that this is due to just 100 trucks).

3. I appreciate that the authors added a limitations section, but I think it that (i) also needs to mention that there is substantial uncertainty about both the VSL and the magnitude of the mortality effects of ambient PM_{2.5} exposure; and (ii) because the uncertainty is not fully captured by the confidence intervals given by the authors, it seems to me that it would be appropriate to list the key limitations in the main text/discussion, as opposed to relegating all of it to the end of the methods section.

(Remarks on code availability)

Version 2:

Reviewer comments:

Reviewer #2

(Remarks to the Author)

The authors have addressed my comments and the additional clarifications and grid emission factors have improved the manuscript.

(Remarks on code availability)

Reviewer Comments:

Responses to Reviewer #1

We thank you for your excellent comments on our paper. We have taken your suggestions seriously and have done our best to address them in the manuscript. Doing so has significantly improved the manuscript. Our point-by-point responses and descriptions of the changes are included below.

1. Since you model trucking corridors across the country, do you account for regional factors that would impact vehicle performance (e.g., grade, temperature)? Do you account for any regional cost differences (diesel prices, electricity prices)?

Response:

Thank you for bringing up the question of regional variation. We do account for the average grade between nodes along corridors in our model of vehicle energy consumption (for both BEV and diesel trucks). This has been clarified in the text. While this will not fully capture smaller uphill and downhill segments, it will capture larger changes in grade that span most or all of a road segment (e.g., going over a mountain range).

Regarding temperature, while we agree that temperature (particularly cold environments) does have an impact on short-term battery performance,^{1,2} this is primarily impactful for passenger EVs and light duty vehicles performing short trips. For trips with several hour long durations, interviews with industry stakeholders revealed that thermal management strategies, and the tendency for batteries to generate heat, substantially reduce the impact cold weather has on battery performance.^{3,4} We also found that HVAC loads were near negligible relative to the power demand for locomotion in heavy-duty trucks, even in high/low temperatures. Ultimately, we determined that battery performance/vehicle energy consumption variation due to local temperature variation was not likely to substantially change our results enough to justify the increase in computational resources needed to include it. However, we acknowledge that future works could improve upon our analysis with an improved thermodynamic model. We have clarified these modeling decisions in lines 620-632 and 914-918 of the methods section in the manuscript, as provided below.

Regarding regional cost differences in diesel and electricity, we do not have reliable future projections for regional differences in diesel prices and so, in the interest of using fairly simple and transparent assumptions, we used a single diesel price for the national analysis. Similarly, prices paid at local charging stations will be dependent on a host of difficult-to-predict factors and they may not correlate well with, for example, wholesale prices. Thus, we have taken

a similar approach in assigning a single value for electricity purchased at charging stations. We decided that the sensitivity analysis (visualized in Figure 2B and 2C) was the best approach to address this variability and uncertainty. We do realize that this decision could be better justified in the text and added lines 133 through 136 to provide more clarity regarding this comment.

Revised Text (Lines 626-6338):

Due to computational limitations, road grade (Z) is simplified to the average road grade between two nodes of the trucking corridor network detailed in the following section. Future models with greater computational resources and increased time resolutions may see improved accuracy in model results by using instantaneous road grades.

While temperature (particularly cold environments) has negative impacts on short-term battery performance,^{59,60} industry interviews^{16,61} highlight the efficacy of thermal management strategies and heat generation of Li-ion batteries in reducing these impacts, especially for long-term operation like long-haul freight transport. Additionally, the minor impact of HVAC loads relative to energy demand from locomotion in heavy-duty freight, the infrequency of extreme temperature events in most regions, and the high computational resources required for an accurate thermodynamic model led us to exclude the impacts of temperature in our behavioral model.

Revised Text (Lines 914-918): Additionally, our study employs a simple battery performance and degradation model that estimates cycling and calendar aging, but factors like regional temperature, driver behavior, road conditions, local traffic flow, etc. could all impact degradation and instantaneous performance. Improved modeling of these factors along with a physio-chemical model of battery performance may improve results but at the cost of increased computational demand.

Revised Text (Lines 133-136): The sensitivity analysis in Fig. 2B and 2C are performed to capture regional and temporal variations in diesel and electricity prices, as well as uncertainty in costs of establishing a nationwide fast-charging network.

In-text Citations:

59. Zeng, L., Hu, Y., Lu, C., Pan, G. & Li, M. Arrhenius Equation-Based Model to Predict Lithium-Ions Batteries' Performance. *JMSE* **10**, 1553 (2022).
60. Yuksel, T., Tamayao, M.-A. M., Hendrickson, C., Azevedo, I. M. L. & Michalek, J. J. Effect of regional grid mix, driving patterns and climate on the comparative carbon footprint of gasoline and plug-in electric vehicles in the United States. *Environmental Research Letters* **11**, 044007 (2016).

16. Wall, J. Interview with John Wall, Former CTO of Cummins. (2023).
61. Moura, S. Interview with Prof. Scott Moura. (2023).

2. Line 187 - *“Human health impacts included in this study are limited to long-haul HDV tailpipe emissions and electricity generator stack emissions due to the limited availability and high uncertainty of modeling human health damages from battery manufacturing and material extraction, which typically occurs outside the US.” I understand why you have not included human health impacts from manufacturing and material extraction, but it may be beneficial to include human health impacts from vehicle operation beyond tailpipe emissions. Brake and tire wear can represent a significant portion of vehicle PM_{2.5} emissions.*

Response:

We thank the reviewer for raising this concern. We agree that changes in brake and tire wear (BTW) PM_{2.5} emissions are important and we have spent substantial time discussing the issue and attempting to seek out reliable data.^{5,6} Unfortunately, at this time we concluded that there is not yet adequate data to include these factors for class 8 trucks, and there is substantial uncertainty regarding differences in BTW emissions between diesel and battery electric heavy duty vehicles (meaning that it is still unclear, directionally, whether incorporating these changes would favor diesel or electric trucks). Specifically, the opposing impacts of regenerative braking and increased vehicle weight for electric trucks pose much uncertainty. Beddows and Harrison⁷ (2020) demonstrates that regenerative braking substantially reduces braking emissions for passenger BEVs while an increase in weight can substantially increase total BTW emissions. Wen et al. (2024)⁷ uses EMFAC data as the basis for their statement that “Brake wear emissions for BEVs are approximately 50% lower than those for ICEVs due to regenerative braking, while tire wear emissions are comparable.” However, Wen et al. acknowledges that EMFAC data is empirical and limited, and does not account for changes in battery-electric vehicle weight, potentially contributing to an underestimation of BTW emissions. While the Beddows and Harrison⁸ model could theoretically be used to determine the cumulative impact from these factors for light-duty vehicles, this model is likely unsuitable for HDVs given their vehicle weight may be 30 to 40 times greater than light-duty vehicles depending on their payloads and designs. At the time of submission there appears to be little consensus on modeling differences in BTW emissions for heavy duty vehicles specifically.⁹

In order to provide clarity for readers on this subject, we specified that our scope excludes BTW emissions and provided further explanation for this decision in lines 483-490. If the reviewer is aware of data sources we have missed, we would welcome suggestions.

Revised Text (Lines 483-490): We assume transportation impacts are negligible and that differences between non-tailpipe emissions for diesel and BE-HDVs are negligible due to the

high uncertainty regarding the opposing impacts of regenerative braking and increased vehicle weight on brake and tire wear (BTW) emissions. While we acknowledge that BTW is a non-negligible contributor to PM_{2.5} emissions,^{42,43} we find that there is high uncertainty and little consensus in current literature on how to appropriately model the BTW emissions specifically for BE-HDVs given these contributing factors.^{44–46} This topic is worthy of further study, and requires more emissions measurements to determine the most realistic assumptions.

In-text Citations:

42. Grigoratos, T. & Martini, G. Brake wear particle emissions: a review. *Environ. Sci. Pollut. Res. Int.* **22**, 2491–2504 (2015).
43. Cai, H., Burnham, A., Wang, M., Hang, W. & Vyas, A. *The GREET Model Expansion for Well-to-Wheels Analysis of Heavy-Duty Vehicles*. (2015).
44. Ravi, V. *et al. FINAL REPORT: LA100 Equity Strategies - Chapter 11: Truck Electrification for Improved Air Quality and Health*. (2023).
45. Beddows, D. C. S. & Harrison, R. M. PM10 and PM2.5 emission factors for non-exhaust particles from road vehicles: Dependence upon vehicle mass and implications for battery electric vehicles. *Atmos. Environ.* **244**, 117886 (2021).
46. Wen, Y. *et al.* Persistent Environmental Injustice due to Brake and Tire Wear Emissions and Heavy-Duty Trucks in Future California Zero-Emission Fleets. *Environ. Sci. Technol.* **58**, 19372–19384 (2024).

3. *Some clarification on the payload cost would be helpful. Line 594 – “Payload costs for Li-ion BE-HDVs represent the cost of performing the excess tonne-miles due to the offset cargo from the additional battery weight relative to diesel HDVs. The average excess tonne-miles is normalized by the payload capacity of an additional Li-ion BE-HDVs and multiplied by the total cost of ownership excluding payload costs.” Is the assumption that excess weight can be shifted to a different vehicle making the same trip? And therefore the total number of vehicles in the diesel and electric scenarios are the same? It may be worth exploring whether a shift to electric trucks might necessitate an increase in the total number of vehicles (and this could be due to driver hour limitations and range limitations as well as cargo weight limitations) and how such an increase would change the results.*

Response: The current methodology does assume that offset cargo will be carried on a separate trip. While this is stated directly in lines 662 through 666, we restated this assumption in

lines 889-897 to improve overall clarity. Additionally, we added specific mention that the human health and climate impacts are scaled by the portion of payload capacity occupied by the excess cargo. This is best visualized in Fig. 4 where there are relatively small “External - GWP” and “External - Human Health” costs in the “Payload” bar. These cost increases account for the private and external costs induced from additional trucks needed, but per your comment they may not perfectly represent the true costs associated with the need for additional truck operation. We acknowledge that there may be additional costs (i.e. greater system wide congestion, increased brake and tire wear impacts, etc.) that our model does not address and have consequently added a discussion of these costs.

Revised Text(Lines 889-897): Payload costs for Li-ion BE-HDVs represent the cost of hauling the excess tonne-miles due to the offset cargo from the additional battery weight relative to diesel HDVs. The average excess tonne-miles is normalized by the payload capacity of an additional Li-ion BE-HDVs and multiplied by the total cost of ownership excluding payload costs.^{14,15} All external costs are also scaled by this cost factor, thus accounting for the system costs induced by an additional BE-HDV operation. Given the scope of our study, we acknowledge that this approach may omit additional external costs associated with the need for a greater number of overall vehicles (i.e., increased congestion, greater BTW emissions, etc.), and suggest that future works further investigate these topics.

In-text Citations:

14. Hunter, C. *et al. Spatial and Temporal Analysis of the Total Cost of Ownership for Class 8 Tractors and Class 4 Parcel Delivery Trucks.* (2021).
15. Burnham, A. *et al. Comprehensive Total Cost of Ownership Quantification for Vehicles with Different Size Classes and Powertrains.* (2021).

4. Based on figures S36 and S38, I'm somewhat concerned that the difference in fuel economy between the electric and diesel vehicle may be slightly overstated. The figures suggest that the long-haul truck is stopping multiple times every 30 minutes, and never surpassing 60 mph. In such a scenario, the electric truck would have a tangible benefit from regenerative braking compared to the diesel vehicle. However, the difference in fuel economy would be lower were the vehicles operating at a more steady rate. And I think the more steady operation may be a more realistic option.

Response: We thank the reviewer for raising the issue of realistic drive cycles. It is true that, coming in and out of urban areas will involve slower speeds and more stopping, while trucks will likely not stop as often once driving on corridors outside of those areas. This is why we opted to use CARB's drive cycle rather than attempting to create (and justify) one for our

own purposes. However, in response to this comment, we performed a comparison of the energy efficiency in terms of kWh per mile of a truck performing the CARB drive cycle used in this study against a drive cycle where the truck was driving a constant 60 mph (which is greater than the average heavy duty truck speed along all major interstate corridors),¹⁰ with the goal of understanding how significant the effect would be given use of regenerative braking. We found there was an average 23% decrease in energy efficiency for BE-HDVs and a 8% decrease in energy efficiency when comparing the simplified drive cycle to the CARB drive cycle. To further explore the differences in results that may arise from a simplified driving behavior and neglecting the impacts of regenerative braking, we added Fig. S46, which visualizes the modeled social costs of diesel and BE-HDVs when operating on the simplified drive cycle. We direct readers to this figure in lines 677-678.

Revised Text (Lines 681-682): Results with a simplified drive cycle (constant 65 mph, or approximately 105 km/h) are presented in Fig. S46.

Additional Figures: Fig. S46. Social costs with a simplified drive cycle (constant 65 mph)

5. The electricity price of \$0.26/kWh seem a bit low to me, just from comparing with the price currently offered at DC fast chargers. And building chargers capable of providing the energy required for long haul trucks will likely be even more expensive than constructing DC fast chargers. However, trucking companies may make agreements with electricity providers to achieve this price, so this might be fine.

Response: As captured in your comment, we agree that there is much uncertainty and variability in the cost of electricity at DC fast chargers, especially considering that the network of chargers we assume is not actually available in the US. While we believe that \$0.26/kWh is the best central estimate of possible prices considering Burnham et al.¹¹ and US EIA,¹² which provide a national average bulk electricity prices for transportation and an informed estimate of the infrastructure costs required to establish this network, we also recognize the true cost of electricity in this scenario remains unknown and can occupy a wide range of possible values. Even though we consider modeling the future costs of electricity and grid infrastructure out-of-scope, we believe the sensitivity analysis in visualized in Fig. 2 is the best method for capturing the high uncertainty in electricity prices in this scenario, as we capture the possibility of charging prices being nearly 100% higher and 300% lower than our central estimate. This sentiment is currently expressed in lines 175 though 182. In order to provide more resources for readers, we've added Fig. S14 and S15 in the SI to illustrate the impacts of varying electricity prices on final results and directed readers to these figures in Lines 182-183. We appreciate your feedback here and hope our change provides greater clarity on the subject.

Revised Text (Lines 182-183): Additionally, Fig. S14 and S15 illustrate the social costs under a variety of electricity prices.

Additional Figures:

Fig. S14. Social costs under a low renewable cost scenario for a variety of years and electricity prices with a 2% discount rate.

Fig. S15. Social costs under a high renewable cost scenario for a variety of years and electricity prices with a 2% discount rate.

6. It may be worth commenting on how these results can be applied to other settings (e.g., in different countries). I'm not suggesting any additional analysis of other locations, but you might discuss which inputs are likely to change or stay the same in other locations, and whether or not that would have a significant impact on the results.

Response: We appreciate this comment and agree that this is an important consideration for future research. Therefore we included a discussion of the value, differences, and challenges related to extending this work to different regions in the “Future Works” section in lines 863 through 870.

Revised Text (Lines 960-966): Although this study focuses on the US, the methods can be extended to other countries and regions, provided accessible methodologies for connecting charging loads to power plant emissions and modeling of air pollution related human health impacts are available. Fully capturing the impacts of Li-ion battery production requires models capable of capturing these global supply chains.⁴¹ Different countries have distinct electricity mixes, population distributions, background concentrations of emissions, and electricity rate structures that may significantly alter the balance of costs and benefits for BE-HDVs.

In-text Citations:

41. Hao, H. *et al.* Impact of transport electrification on critical metal sustainability with a focus on the heavy-duty segment. *Nat. Commun.* **10**, 5398 (2019).

7. Minor comments and suggestions

-Small typo in line 81 (“explores”)

-The text and the figure captions skip from figure 2 to figure 4

-Citations 3 and 14 appear to be the same

-I'm assuming the 100-year GWP is used for the CO₂ equivalent of other GHGs. It might be good to state that explicitly somewhere in the methods section.

-In figure 5, it would be nice to have the years (2025 and 2035) for panels A and B shown directly on the figure.

Response: Thank you for the minor comments and suggestions. We appreciate you catching these points and we have made the appropriate corrections. Additionally, we have clarified that we are using the AR6 100-year GWP in lines 209 through 212.

Revised Text (Lines 209-212): The SCCs of \$212/tonne CO_{2eq} and \$248/tonne CO_{2eq} are used to determine damages associated with GWP (using the Intergovernmental Panel on Climate Change's 100-year values in the Assessment Report 6) in the 2025 and 2035 scenarios respectively (SCC's visualized in Fig. S16).²²

In-text Citations:

22. US EPA. *Supplementary Material for the Regulatory Impact Analysis for the Final Rulemaking, "Standards of Performance for New, Reconstructed, and Modified Sources and Emissions Guidelines for Existing Sources: Oil and Natural Gas Sector Climate Review" - EPA Report on the Social Cost of Greenhouse Gases: Estimates Incorporating Recent Scientific Advances.* (2023).

Reviewer #1 Response References:

1. Zeng, L., Hu, Y., Lu, C., Pan, G. & Li, M. Arrhenius Equation-Based Model to Predict Lithium-Ions Batteries' Performance. *JMSE* **10**, 1553 (2022).
2. Yuksel, T., Tamayao, M.-A. M., Hendrickson, C., Azevedo, I. M. L. & Michalek, J. J. Effect of regional grid mix, driving patterns and climate on the comparative carbon footprint of gasoline and plug-in electric vehicles in the United States. *Environmental Research Letters* **11**, 044007 (2016).
3. Moura, S. Interview with Prof. Scott Moura. (2023).
4. Wall, J. Interview with John Wall, Former CTO of Cummins. (2023).
5. Grigoratos, T. & Martini, G. Brake wear particle emissions: a review. *Environ. Sci. Pollut. Res. Int.* **22**, 2491–2504 (2015).
6. Cai, H., Burnham, A., Wang, M., Hang, W. & Vyas, A. *The GREET Model Expansion for Well-to-Wheels Analysis of Heavy-Duty Vehicles.* (Argonne National Laboratory, 2015).

7. Wen, Y., Yu, Q., He, B. Y., Ma, J., Zhang, S., Wu, Y. & Zhu, Y. Persistent Environmental Injustice due to Brake and Tire Wear Emissions and Heavy-Duty Trucks in Future California Zero-Emission Fleets. *Environ. Sci. Technol.* **58**, 19372-19384 (2024).
8. Beddows, D. C. S. & Harrison, R. M. PM10 and PM2.5 emission factors for non-exhaust particles from road vehicles: Dependence upon vehicle mass and implications for battery electric vehicles. *Atmos. Environ.* **244**, 117886 (2021).
9. Ravi, V., Li, Y., Heath, G., Marroquin, I., Day, M. & Walzberg, J. *FINAL REPORT: LA100 Equity Strategies - Chapter 11: Truck Electrification for Improved Air Quality and Health*. (National Renewable Energy Laboratory, 2023).
10. U.S. DOE. Fact #671: April 18, 2011 Average Truck Speeds. *U.S. Department of Energy* at <[https://www.energy.gov/eere/vehicles/fact-671-april-18-2011-average-truck-speeds#:~:text=The%20Federal%20Highway%20Administration%20studies,mph%20\(I%2D81\)>](https://www.energy.gov/eere/vehicles/fact-671-april-18-2011-average-truck-speeds#:~:text=The%20Federal%20Highway%20Administration%20studies,mph%20(I%2D81)>)>
11. Burnham, A., Dufek, E. J., Stephens, T., Francfort, J., Michelbacher, C., Carlson, R. B., Zhang, J., Vijayagopal, R., Dias, F., Mohanpurkar, M., Scoffield, D., Hardy, K., Shirk, M., Hovsopian, R., Ahmed, S., Bloom, I., Jansen, A. N., Keyser, M., Kreuzer, C., Markel, A., Meintz, A., Pesaran, A. & Tanim, T. R. Enabling fast charging – Infrastructure and economic considerations. *J. Power Sources* **367**, 237–249 (2017).
12. US Energy Information Administration. *Annual Energy Outlook 2023*. (US Energy Information Administration, 2023).

Responses to Reviewer #2

We thank you for your excellent comments on our paper, particularly related to our external cost calculations. We have taken your suggestions seriously and have done our best to address them in the manuscript. Doing so has significantly improved the manuscript. Our point-by-point responses and descriptions of the changes are included below.

1. The paper provides little of the underlying data and assumptions that would be necessary to fully appreciate its results. It does not provide, for example, emission factors for the electric generating units; the concentration-response function used to link PM_{2.5} concentrations to mortality; the value per statistical life (VSL) used to monetize those impacts. It also does not provide the base year for any of these (e.g., what VSL applies to 2025 and 2035 and how was it VSL adjusted)

Response: We thank the reviewer for highlighting this concern. Our article covers a lot of ground, including the truck fleet model, cost model for both electric and diesel trucks, electricity grid model, and human health impact assessment. In some cases, we have pointed to underlying documentation where these values exist, but we are happy to expand the documentation provided directly in our article and SI. In response to this request, we have provided the concentration-response function and the value of a statistical life in lines 751 through 809 in the “Grid Emissions and External Costs” section. With regards to the emission factors for EGUs, the values used are specific to individual generators (from in Jenn et al.¹) and have been provided by Dr. Alan Jenn. We are unable to publish these emission factors without permissions from Dr. Alan Jenn, but we have directed readers to Jenn et al.¹ in lines 721 through 723.

Revised Text (Lines 755-809): Annual mortality from PM_{2.5} emissions is modeled using the concentration-response functions from Krewski et al.⁵³ and Lepeule et al.,⁵⁴ provided in Equations 7 and 8 respectively, where $Deaths_K$ and $Deaths_L$ represent the number of deaths using the Krewski and Lepeule equations respectively. Additionally, $PM_{2.5}$ represents emissions in kg, $Population$ represents the total population per grid cell, $Mortality\ Rate$ represents the year 2005 baseline overall population mortality rate in deaths per year per 100,000 people and mortality rate increases by 6% and 14% for every 10 $\mu\text{g}/\text{m}^3$ PM_{2.5} for the Krewski et al.⁵³ and Lepeule et al.⁵⁴ equations respectively. Tessum et al.,²⁴ Goodkind et al.²⁵ and Thind et al.⁷¹ provide greater detail on how these concentration-response functions are integrated into the ISRM. When modeling human health damages, a \$11.51 million value of a statistical life (VSL)⁷⁷ is applied to the mean deaths from the Krewski et al.⁵³ and Lepeule et al.⁵⁴ equations. Note that this VSL is used for results in 2025 and 2035 following procedures by Goforth and Nock⁷⁸ and Jackson et al.⁷⁹ to limit the uncertainty of future mortality valuations.

$$Eq\ 7: Deaths_K = \left(e^{\left(\frac{\log(1.06)}{10} * PM_{2.5} \right)} - 1 \right) * \frac{Population * Mortality\ Rate}{100,000}$$

$$Eq\ 8: Deaths_L = \left(e^{\left(\frac{\log(1.14)}{10} * PM_{2.5} \right)} - 1 \right) * \frac{Population * Mortality\ Rate}{100,000}$$

Revised Text (Lines 721-723): The emitted GHGs and criteria air pollutants at the responding marginal generators are then estimated using generator-specific emission factors from the Grid Optimized Operation Dispatch Model developed by Jenn et al.⁶⁹

In-Text Citations:

53. Krewski, D. *et al.* Extended follow-up and spatial analysis of the American Cancer Society study linking particulate air pollution and mortality. *Res Rep Health Eff Inst* 5–114; discussion 115 (2009).
54. Lepeule, J., Laden, F., Dockery, D. & Schwartz, J. Chronic exposure to fine particles and mortality: an extended follow-up of the Harvard Six Cities study from 1974 to 2009. *Environ. Health Perspect.* **120**, 965–970 (2012).
24. Tessum, C. W. *et al.* Inequity in consumption of goods and services adds to racial-ethnic disparities in air pollution exposure. *Proc Natl Acad Sci USA* **116**, 6001–6006 (2019).
25. Goodkind, A. L., Tessum, C. W., Coggins, J. S., Hill, J. D. & Marshall, J. D. Fine-scale damage estimates of particulate matter air pollution reveal opportunities for location-specific mitigation of emissions. *Proc Natl Acad Sci USA* **116**, 8775–8780 (2019).
71. Thind, M. P. S., Tessum, C. W., Azevedo, I. L. & Marshall, J. D. Fine Particulate Air Pollution from Electricity Generation in the US: Health Impacts by Race, Income, and Geography. *Environ. Sci. Technol.* **53**, 14010–14019 (2019).
77. US EPA. Mortality Risk Valuation. *Mortality Risk Valuation* <https://www.epa.gov/environmental-economics/mortality-risk-valuation> (2025).
78. Goforth, T. & Nock, D. Air pollution disparities and equality assessments of US national decarbonization strategies. *Nat. Commun.* **13**, 7488 (2022).
79. Jackson, C. M., Holloway, T. & Tessum, C. W. City-scale analysis of annual ambient PM_{2.5} source contributions with the InMAP reduced-complexity air quality model: a case study of Madison, Wisconsin. *Environmental Research: Infrastructure and Sustainability* **3**, 015002 (2023).

70. Jenn, A., Clark-Sutton, K., Gallaher, M. & Petrusa, J. Environmental impacts of extreme fast charging. *Environmental Research Letters* **15**, 094060 (2020).

2. Emission factors for PM2.5 for diesel trucks seem extraordinarily high, and NOx emission factors also seem high. NOx and PM2.5 emissions are the pollutants that will cause almost 100% of the human health of the damages of the 2025 diesel trucks – the contribution from SO2 would be negligible (In the case of the 2035 diesel truck it is possible that NH3 emissions reported by the authors also contribute some).

On PM2.5:

Table S9 provides 2.20 g PM2.5/kg-fuel for 2025 and 0.30 g PM2.5/kg-fuel in 2035. Using the fuel economies provided in Table S9 and assuming a density of 0.84 kg/L for diesel, this is equivalent to 1.30 g PM2.5/mi in 2025 and 0.18 g PM2.5/mi in 2035. The resulting value for 2025 is about 450 times higher (and even the 2035 value is about 60 times higher) than the 0.0028 g/mi PM2.5 (lifetime-weighted average) reported for GREET for a model year 2010 class 8 truck by Cai et al. (2015) for an earlier version of GREET [see Table 18, p. 42 in Cai et al. 2015]. Cai et al. also report 0.0032 g PM2.5/mi in Tire and brake wear emissions. Even if the authors also meant include those (although the title of table S9 says “tailpipe”), the author’s value for 2025 is still 200 times higher than total PM2.5 emissions in Cai et al. (the value for 2035 would be about 30 times higher)

On NOx:

A similar comparison for NOx suggests that the authors’ values for 2025 are 30 times higher than the 0.2755 g/mile reported in Table 18 from Cai et al. (2015)

The values cited above are from an earlier version of GREET, as I was not able to access the GREET website at this time to provide with a citation for more a recent version of GREET. I acknowledge that especially NOx emission rates may have been revised and substantially increased in more recent versions of the model.

Another comparison can be done with fleet-average values in 2025 reported for heavy duty vehicles (other than buses) by the Bureau of Transportation Statistics (BTS) (2025) based on EPA MOVES4. BTS (2025) reports:

PM2.5: 0.054 g/mi (exhaust) (also 0.031 g/mi in brake wear and 0.003 g/mi in tire wear). NOx: 2.942 g/mi.

The current paper’s values are about 24 times higher for PM2.5 and 3 times higher for NOx. I note, however, that this comparison with fleet average emission factors in 2025 is not adequate since fleet-wide emissions for PM2.5 and NOx are much higher than for new model years as they are dominated by a small fraction of pre-2006 MY vehicles circulating which emit at very high rates. This is evidenced by the decreasing fleet-wide averages estimated by BTS through 2030, even in the absence of new major regulation. It seems difficult to understand how the authors estimate values higher than those fleet-wide averages.

Response:

We sincerely thank the reviewer for their comment and this has prompted us to make some changes. We have organized our response into two sections: (1) Identifying the error in Table S9 (revised to Table S8), and (2) Comparing the correct EFs to literature.

(1) Identifying the error in Table S9.

After reviewing our methodology of this work and the works that have informed it, we have identified that Table S9 in its current form contains two typos, one that we are responsible for, and one that is attributable to the source of these EFs, Tong et al.² (2021). In short, our analysis used correct emission factors, but Table S9 contained two typos, which have been corrected such that it is consistent with the factors we actually used in our modeling.

The first typo in Table S9 was that the Pollution Control Technology for the Diesel Truck 2025 should be “DPF + SCR” instead of “DPF.” In our model, the emission factors we use reflect trucks with diesel particulate filter (DPF) and selective catalytic reduction (SCR) technology.

The second typo was actually carried over from the source of these EFs, Tong et al.,² which reports these emission factors in their Table S8. In both cases (our study and Tong et al.), the actual emission factors used in the analysis are correct but the tables documenting these emission factors contained several errors. See a marked-up table below indicating the typos in the Tong et al. table:

[editorial note: third party material is redacted]

Here we see that the EF for PM_{2.5} and NO_x is 2.2 and 14.2 g/kg fuel respectively for MY2010-2018, while the EFs for incremental and advanced designs are 0.3 and 5.2 for PM_{2.5} and NO_x respectively. However, when we look at the results of Tong et al. presented in Figure 2a, MY 2010-2018 vehicles only have marginal differences in total PM_{2.5} and NO_x emissions relative to incremental and advanced designs. Thus we identified that the MY2010-2018 EF for PM_{2.5} used in Tong et al.² is indeed 0.3 g/kg, and the EF for NO_x is 5.2 (and the EFs for NH₃ and CH₄ are 0.18 and 0.67 respectively), and that the marginal difference in emissions in Figure 2a is attributable to the improved fuel economy of the incremental and advanced designs. We worked with the primary author of Tong et al.² (who is a contributing author on this paper) to review the code and methodology of Tong et al.,² and we were able to confirm that these are the EFs used to perform their analysis. Additionally, we confirmed that our analysis, which was built from the preceding Tong et al.,² does indeed use the correct emission factors.

Thus, we have corrected Table S9 (revised to Table S8) to display the actual EFs used in our study: 0.3 g PM_{2.5} per kg fuel, 5.2 g NO_x per kg fuel, 0.18 g NH₃ per kg fuel, and 0.67 g CH₄ per kg fuel. We will also promptly issue a correction for Tong et al.² explaining the typo and noting that the error in documentation did not actually impact the analysis. We sincerely thank the reviewer for their thorough work and identifying this inconsistency.

(2) Comparing the correct EFs to literature.

Again, we thank the reviewer for performing a thorough comparison of our emissions factors. The corrected PM_{2.5} EF is now equivalent to 0.18 g / mile and the NO_x EF is 3.06 g / mile. When comparing this to the most recent and relevant literature values referenced by the reviewer (EPA MOVES4 2025)³, it is now clear that our PM_{2.5} EF are only 3.3 times larger and our NO_x EF is nearly equivalent. We acknowledge that the 3 times increase in PM_{2.5} emissions is not negligible, but we are confident in the methods in Preble et al. 2019 and Tong et al. 2021 used to determine them. Specifically, Preble et al. measures BC emissions directly from HDV tailpipe plumes and Tong et al. employs the ratio of BC to PM_{2.5} by vehicle model year from Constantini et al. 2016 to determine total PM_{2.5} emissions. This approach reflects on-road emissions and captures the current impacts of mal-maintenance and tampering that have not been updated in EPA MOVES4 since 2007 (page 60).³ Additionally, EPA MOVES³ emission rates are determined through drive-cycle testing which may not accurately reflect real world driving conditions.

Additionally, as the reviewer demonstrated, PM_{2.5} EFs vary considerably throughout the literature. In particular, while the comparison of our work to Cai et al.⁴ is appreciated and was vital for identifying our critical typos in Table S9, we find that this specific comparison should not be applied to our corrected EFs, especially considering that the EPA MOVES4³ PM_{2.5} EF is over 23 times greater than the EF in Cai et al.⁴ This high level of variability throughout literature

can be attributed to improving methodologies and the rapidly advancing state of knowledge on these topics. After reviewing our methods and consulting with experts, we confirmed that our methodology accurately reflects heavy-duty vehicle tailpipe emissions for current, real world driving conditions to the best of our ability.

Revised Table: Table S8. Tailpipe Emission Factors of Long-haul Diesel Trucks by Scenario

3. Power plant damages also seem much higher than the literature.

For example, Dedoussi et al. (2020) estimates roughly 7,000 PM_{2.5}-attributable deaths due to grid emissions in 2018. For a generation of about 4,200 TWh in that year

(<https://www.eia.gov/todayinenergy/detail.php?id=38572>) and a VSL of roughly 12 million (US DHHS, 2021), this would be equivalent to 2 cents/kWh. Choma et al. (2024) also report a similar value of 2.2 cents/kWh for 2018 using InMAP – which further decreases over time. Furthermore, SO₂ emission rates per kWh dropped by 45% between 2018 and 2023, and NO_x emission rates by 33% (U.S. EPA, 2024a), so one would expect closer to 1 cent/kWh for 2025. The authors, on the other hand, report 5.8 cents/kWh for low RE and 6.6 cents/kWh for high RE (Table S1). Part of the difference could be explained by their choice of a marginal model, though it still seems difficult to reconcile damages per kWh that are an order of magnitude higher than suggested in the literature. Perhaps even more importantly, the electric truck would not consume the marginal electricity in 2025 but rather the marginal electricity at the time it is charged – over the next 15 years or so. This would likely decrease their health damages substantially if forecasted decreases in grid emission rates are realized.

Response:

We thank the reviewer for highlighting this concern. We note that the reviewer's intuition on the difference between the impacts of marginal and average generation models is indeed responsible for our elevated power plant damages relative to literature. In fact, the reviewer's calculations agree with literature that explores these differences. Gagnon and Cole⁵ employ 3 models alongside observed data to find that short-run marginal emission factors (EFs) are roughly 2-3 times higher than average EFs, which exactly describes the difference in values between Dedoussi et al. 2020 and Choma et al. 2024 when compared to those used in this manuscript. Additionally, given the size and nature of the truck charging loads, the decision to model their impacts via a short-run marginal model is justified considering the precedent established by literature.^{1,5-12}

The reviewer also brought to light another valid concern regarding how emission factors change over time. The reviewer correctly highlighted that average emission factors are

decreasing over time; this behavior does not necessarily extend to marginal emissions, as discussed by Holland et al.¹³ and Bruchon et al.,¹⁴ which in some cases have increased over time. For example, at times when variable renewable energy generation is high, it can actually cause coal plants to ramp down (this has been observed already), meaning that coal-fired power generation is on the margin and increases in load during those hours result in coal generation actually increasing. Therefore, while the reviewer correctly decreases the average emissions in their calculation, this method does not extend to marginal emissions used in this manuscript. Overall, we find that the modeling choices and results in this manuscript are actually supported by the differences in average and short-run marginal emissions models raised by the reviewer's astute literature comparisons and calculations.

Additionally, the reviewer raises a fair concern that the battery electric HDVs will be using marginal electricity over the full lifetime of a HDV. However, this study only models the lifespan of a truck operating in long-haul freight, of which most vehicles only spend 4 years.¹⁵⁻¹⁷ (line 115) The shorter operational lifetime and scope and relative constant emissions associated with marginal emissions both explain the differences in emissions and support our modeling decisions.

We thank the reviewer for raising the importance of our modeling choice regarding the emissions model and have improved our manuscript in response. Specifically, we added Discussion S1 on the measured differences between average and marginal generation models, considering the points raised by the reviewer and directing readers to literature highlighted in our response. We direct readers to this discussion in lines 691-698.

Revised Text (Discussion S1):

Discussion S1

This study employs a marginal short-run generation model when estimating generator response, following the precedent set throughout literature when evaluating electricity impacts associated with small quantities of electric vehicles inducing relatively small changes in generator behavior.¹⁻¹⁰ In this study, the induced generation demand from BE-HDVs relative to the total quantity of generation nationwide is near negligible and will rely on marginal generation resources (i.e. not baseload), allowing for the use of a marginal short-run generation model.

For studies that examine dramatic changes to electricity demand or the impacts associated with average electricity demand, a short-run marginal generational model may mischaracterize generation impacts. Gagnon and Cole⁹ employ 3 models alongside observed data to find that short-run marginal emission factors (EFs) are roughly 2-3 times higher than average EFs and long-run marginal EFs. This can largely be attributed to the increasing tendency to use coal

generation and reliance on less efficient generators to meet marginal electricity demand, resulting in marginal EFs that have been slightly increasing over time despite average emissions decreasing.^{11,12} Importantly, Holland et al. finds that EFs of marginal generators are substantially higher than EFs of average generators of the same fuel type due to differences in efficiency attributable to lower utilization rates.

Future studies regarding the electrification should carefully consider whether a marginal short-run generation model accurately characterizes their load type and corresponding generator response, as well as whether emissions associated with this modeling choice will substantially vary over time.

Revised Text (Lines 695-702): The hourly electricity power demand at charging nodes is used to determine the use phase contributions to GWP and human health burden from Li-ion BE-HDV operation. We employ the short-run marginal generator electricity grid model outlined in McNeil et al. 2024⁷ and McNeil et al. 2025²¹ to determine marginal generator type and location responding to the hourly electricity demand at each node. This model considers power flow between regions at an hourly resolution and how the grid evolves over the next several decades due to renewable energy sources taking over much of the marginal generation in many regions. Justification and implications of using a marginal short-run generator model are provided in Discussion S1.

In-text Citations:

1. Siler-Evans, K., Azevedo, I. L. & Morgan, M. G. Marginal emissions factors for the U.S. electricity system. *Environ. Sci. Technol.* **46**, 4742–4748 (2012).
2. Graff Zivin, J. S., Kotchen, M. J. & Mansur, E. T. Spatial and temporal heterogeneity of marginal emissions: Implications for electric cars and other electricity-shifting policies. *J. Econ. Behav. Organ.* **107**, 248–268 (2014).
3. Archsmith, J., Kendall, A. & Rapson, D. From cradle to junkyard: assessing the life cycle greenhouse gas benefits of electric vehicles. *Research in Transportation Economics* **52**, 72–90 (2015).
4. Hoehne, C. G. & Chester, M. V. Optimizing plug-in electric vehicle and vehicle-to-grid charge scheduling to minimize carbon emissions. *Energy* **115**, 646–657 (2016).
5. Jenn, A., Clark-Sutton, K., Gallaher, M. & Petrusa, J. Environmental impacts of extreme fast charging. *Environmental Research Letters* **15**, 094060 (2020).

6. Tong, F., Jenn, A., Wolfson, D., Scown, C. D. & Auffhammer, M. Health and Climate Impacts from Long-Haul Truck Electrification. *Environ. Sci. Technol.* **55**, 8514–8523 (2021).
7. McNeil, W. H., Tong, F., Harley, R. A., Auffhammer, M. & Scown, C. D. Corridor-Level Impacts of Battery-Electric Heavy-Duty Trucks and the Effects of Policy in the United States. *Environ. Sci. Technol.* **58**, 33–42 (2024).
8. McNeil, W. H. *et al.* Impact of truck electrification on air pollution disparities in the United States. *Nat. Sustain.* (2025) doi:10.1038/s41893-025-01515-x.
9. Gagnon, P. & Cole, W. Planning for the evolution of the electric grid with a long-run marginal emission rate. *iScience* **25**, 103915 (2022).
10. Ryan, N. A., Johnson, J. X. & Keoleian, G. A. Comparative assessment of models and methods to calculate grid electricity emissions. *Environ. Sci. Technol.* **50**, 8937–8953 (2016).
11. Holland, S. P., Kotchen, M. J., Mansur, E. T. & Yates, A. J. Why marginal CO₂ emissions are not decreasing for US electricity: Estimates and implications for climate policy. *Proc Natl Acad Sci USA* **119**, (2022).
12. Bruchon, M., Chen, Z. L. & Michalek, J. Cleaning up while Changing Gears: The Role of Battery Design, Fossil Fuel Power Plants, and Vehicle Policy for Reducing Emissions in the Transition to Electric Vehicles. *Environ. Sci. Technol.* **58**, 3787–3799 (2024).
7. McNeil, W. H. *et al.* Impact of truck electrification on air pollution disparities in the United States. *Nat. Sustain.* (2025) doi:10.1038/s41893-025-01515-x.
21. McNeil, W. H., Tong, F., Harley, R. A., Auffhammer, M. & Scown, C. D. Corridor-Level Impacts of Battery-Electric Heavy-Duty Trucks and the Effects of Policy in the United States. *Environ. Sci. Technol.* **58**, 33–42 (2024).

4. The InMAP model used by the authors calculates damages using 2011 data – not just on emissions but also e.g. on the VSL (Goodkind *et al.*, 2019) and I believe on demographics and mortality rates as well. The authors should explain how they applied 2011 InMAP values to 2025 and what adjustments, if any, were made.

Response: We appreciate the reviewer's comment and have indeed added specific mention of the updated VSL value we use (line 801-803). With regard to background concentration and demographics, we followed the procedures established by Goforth and Nock¹⁸ and Jackson et al.,¹⁹ which project results out to 2050 while maintaining current background concentrations and demographics. We acknowledge that updating these factors may alter results; however, it would be an immense undertaking to completely recreate the data matrix for the ISRM for present concentrations and demographics. We have added discussion of this limitation in the added Limitations section on lines 918 through 925.

Revised Text (Lines 801-803): When modeling human health damages, a \$11.51 million value of a statistical life (VSL)⁷⁷ is applied to the mean deaths from the Krewski et al.⁵³ and Lepeule et al.⁵⁴ equations.

Revised Text (Lines 920-927): The ISRM used to model air pollution impacts in this study is also designed for computational efficiency but it makes several simplifying assumptions, as discussed in greater detail in McNeil et al.⁷ and Tessum et al.²⁴ Additionally, the ISRM relies on background concentration and demographic data that is not updated to the current year or future years. Updating this data may improve results but is a tremendous undertaking for a study not primarily focused on background concentrations and demographics; thus it is not standard practice throughout literature.^{7,21,78,79} Development of standard scenarios for future demographic changes and background air pollutant concentrations would be of great value to the research community.

In-text Citations:

77. US EPA. Mortality Risk Valuation. *Mortality Risk Valuation* <https://www.epa.gov/environmental-economics/mortality-risk-valuation> (2025).
53. Krewski, D. *et al.* Extended follow-up and spatial analysis of the American Cancer Society study linking particulate air pollution and mortality. *Res Rep Health Eff Inst* 5–114; discussion 115 (2009).
54. Lepeule, J., Laden, F., Dockery, D. & Schwartz, J. Chronic exposure to fine particles and mortality: an extended follow-up of the Harvard Six Cities study from 1974 to 2009. *Environ. Health Perspect.* **120**, 965–970 (2012).
7. McNeil, W. H. *et al.* Impact of truck electrification on air pollution disparities in the United States. *Nat. Sustain.* (2025) doi:10.1038/s41893-025-01515-x.

24. Tessum, C. W. *et al.* Inequity in consumption of goods and services adds to racial-ethnic disparities in air pollution exposure. *Proc Natl Acad Sci USA* **116**, 6001–6006 (2019).
21. McNeil, W. H., Tong, F., Harley, R. A., Auffhammer, M. & Scown, C. D. Corridor-Level Impacts of Battery-Electric Heavy-Duty Trucks and the Effects of Policy in the United States. *Environ. Sci. Technol.* **58**, 33–42 (2024).
78. Goforth, T. & Nock, D. Air pollution disparities and equality assessments of US national decarbonization strategies. *Nat. Commun.* **13**, 7488 (2022).
79. Jackson, C. M., Holloway, T. & Tessum, C. W. City-scale analysis of annual ambient PM_{2.5} source contributions with the InMAP reduced-complexity air quality model: a case study of Madison, Wisconsin. *Environmental Research: Infrastructure and Sustainability* **3**, 015002 (2023).

5. The authors compare their results with the National Academies report by Cohon *et al.* (The authors' reference number 28) (National Research Council, 2010). However, the report by Cohon *et al.* uses 2005 emissions data (as well as a much simpler air pollution model) which renders the comparisons difficult to understand. For example, in Table 2-11 (p. 97) Cohon *et al.* mentions an average of 12 lb. SO₂ (a value of 10.1 is also given in p. 108 of Cohon *et al.*) and 4.1 lb. NO_x per MWh for coal plants in 2005. These values are several times higher than what the US EPA (2024a) estimates for the average coal plant in 2023 (1.7 lb SO₂/MWh and 1.2 lb NO_x/MWh) – not to mention that the coal share of generation in 2023 is much lower than it was in 2005.

The difference is perhaps even larger for vehicles. Table D-9, Appendix D, p. 462 in Cohon *et al.* give 0.36 g PM_{2.5}/mi and 12.89 g NO_x/mi for a Heavy Duty Diesel Vehicle Class 8a and 0.36 g PM_{2.5}/mi and 15.10 g NO_x/mi for a Class 8b HDDV. These values are about 1-2 orders of magnitude higher than today (see values in point 3. Above).

In this case, while it seems difficult to fully appreciate any comparisons of the present study by Cohon *et al.*, it is surprising that they yield similar externalities per mile despite drastic changes in emissions over the past 20 years.

Response:

We thank the reviewer for their detailed comments in this area. We would like to clarify that the goal of this literature review is to validate our results against a body of literature with a variety of methods. Cohon *et al.*²⁰ was selected as a point of comparison given its similar scope and high impact throughout this research and policy space, not because of methodological

similarities. However, despite the methodological differences, we find that the similarity in our results can be explained by several factors and demonstrates the value of such a comparison.

Similar to comment three, the incongruity in power plant emissions pointed out by the reviewer can be explained by the comparison of average generation values relative to our marginal generation model. We direct the reviewer to our response to comment three for further evidence and explanation of this phenomenon.

Given the differences in power plant emissions, we agree with the reviewer that it is difficult to directly compare our results with average emission metrics, like those presented in Cohon et al.²⁰ Thus, we improved the “Literature Review and Comparison” section (lines 573-574) to direct readers to Discussion S1 to explain the following points: (1) we use a marginal model (and are justified in using a marginal model) while past literature tends to present average values; (2) average values have decreased overtime while marginal values have remained relatively constant; (3) even though average values have decreased substantially, marginal generators tend to have substantially greater damages for a variety of factors illustrated in Holland et al.¹³ and Bruchon et al.¹⁴ Critically, this last point remains true even when comparing average and marginal generators of the same generator type (i.e. comparing average coal damages to marginal coal generator damages as done by the reviewer) due to decreased efficiencies with lower utilization rates as pointed out by Holland et al. 2022. Again, the reviewer’s comment validates these points and supports the narrative reached in our revised “Literature Review and Comparison.”

As per vehicle emissions, we find that the effects of Cohon et al.’s²⁰ greater emission factors are largely negated by the decreased population size at the time of analysis and use of a simpler air pollution model (APEEP). We have revised our comparison to include discussion of these points in lines 546-553. Once again, we thank the reviewer for helping to improve and clarify the narrative of this study.

Revised Text (576-577): This comparison is consistent with our use of a marginal short run marginal generator, as per Discussion S1.

Revised Text (549-556): If we ignore the climate damages, which are highly variable depending on the SCC used, and compare the human health damages of the diesel HDVs in this study (\$0.50 per VMT) to those in Cohon et al.²⁶ whose boundaries are most similar to this study’s, we find that our results are central to their non-forecasted estimates (between -\$0.008 and \$0.94 per VMT in 2005) despite Cohon et al. using higher emission factors representative of older pollution control technologies in HDVs. We find that a decreased population size at the time of analysis and their use of a simpler air pollution model counteract their elevated damages, resulting in similar net damages.

In-text Citations:

26. Cohon, J. L. *et al. Hidden costs of energy: unpriced consequences of energy production and use.* (National Academies Press, 2010). doi:10.17226/12794.

6. I appreciate the authors' attempt to quantify the uncertainty about several components of the model; however, uncertainty about the externalities is not discussed. Uncertainty about several aspects included in the calculation of the external costs (e.g. the dose-response between PM_{2.5} and mortality, especially for diesel vehicles) may be one of the largest contributors to the uncertainty in their results. While it may be difficult to provide with numerical values for this uncertainty, this should be at least discussed. The uncertainty in the Social Cost of Carbon is also about an order of magnitude (Rennert *et al.*, 2022) and does not seem to be incorporated by the authors.

Response: We thank the reviewer for highlighting the importance of including this discussion. To address this, we provided a discussion of the uncertainty of external costs in the “Grid Emissions and External Costs” section on lines 805-820, with emphasis on the dose-response curve, VSL, and social cost of carbon. We note that given the scope of this paper we are unable to more thoroughly discuss the intricacies of uncertainty tied to the biological response to PM_{2.5}, the philosophical/economic quandaries surrounding the valuation of a life, and the many modes of climate change may impact humanity (i.e. sea-level rise, agricultural response, zoological behavioral changes, etc.); However, our discussion provides sources that may direct readers to discussions on these topics, which we are unable to address in this study.

Revised Text (Lines 806-822):

$$Eq\ 7: Deaths_K = \left(e^{\left(\frac{\log(1.06)}{10} * PM_{2.5} \right)} - 1 \right) * \frac{Population * Mortality\ Rate}{100,000}$$

$$Eq\ 8: Deaths_L = \left(e^{\left(\frac{\log(1.14)}{10} * PM_{2.5} \right)} - 1 \right) * \frac{Population * Mortality\ Rate}{100,000}$$

The above relationship between PM_{2.5} and mortality may be a substantial source of uncaptured uncertainty, given regional and temporal variations in the vulnerability of local populations^{80,81} as well as potential methodological errors tied to dose-response curve development.⁸² These drivers of epistemic and aleatory uncertainty are estimated to contribute around +/-3% change in deaths for a given PM_{2.5} concentration.^{80,81} Additional sources of uncertainty are associated with the use of InMAP and are further discussed in the “Limitations” section. Furthermore, there is uncertainty surrounding the VSL due to imperfect measures of risk

and value used to estimate it, along with the inherent sampling bias present in complex statistical analyses.^{83,84} A recent literature review demonstrated that the estimated VSL may vary by as much as +/-50% across different studies.⁸⁴ Further, estimation of the social cost of carbon also relies on complex climate models and damage functions for the many modes of climate impacts that have multiple sources of uncertainty. Recent estimates show uncertainty has a significant impact on the SCC estimate depending on the discount rate used during calculation.^{22,51}

In-text Citations:

80. Vodonos, A., Awad, Y. A. & Schwartz, J. The concentration-response between long-term PM2.5 exposure and mortality; A meta-regression approach. *Environ. Res.* **166**, 677–689 (2018).
81. Schwartz, J., Laden, F. & Zanobetti, A. The concentration-response relation between PM(2.5) and daily deaths. *Environ. Health Perspect.* **110**, 1025–1029 (2002).
82. Srinivasan, B. & Lloyd, M. D. Quantitation and error measurements in dose–response curves. *J. Med. Chem.* **68**, 2052–2056 (2025).
83. Viscusi, W. K. & Aldy, J. E. *The Value of a Statistical Life: A Critical Review of Market Estimates Throughout the World.* (2003).
84. Kearsley, A. *HHS Standard Values for Regulatory Analysis, 2024.* (2024).
22. US EPA. *Supplementary Material for the Regulatory Impact Analysis for the Final Rulemaking, “Standards of Performance for New, Reconstructed, and Modified Sources and Emissions Guidelines for Existing Sources: Oil and Natural Gas Sector Climate Review” - EPA Report on the Social Cost of Greenhouse Gases: Estimates Incorporating Recent Scientific Advances.* (2023).
51. Rennert, K. *et al.* Comprehensive evidence implies a higher social cost of CO2. *Nature* **610**, 687–692 (2022).

7. The authors should explain better the rationale for discounting different types of impacts with different discount rates. As one example, Lines 184-186 mention a 5% discount rate for health and 2% for the social cost of carbon. However, about half of the social cost of carbon is thought to be increased mortality from extreme heat (Rennert *et al.*, 2022) so it is unclear why the two impacts merit different discount rates. The authors also do not explain whether they account for a cessation lag between changes in PM2.5 exposure and mortality effects (e.g., US EPA 2024b).

Response: Thank you for this comment. We struggled with this approach initially, as there is no clear guidance or common practice. We used a conservative approach here, as the higher discount rate on health damages, drives down the external cost. We have updated all our results and figures to use a 2% across the board Social Discount Rate, which has been advocated for in the literature.²¹ We explain this choice in lines 212 through 229.

Revised Text (Lines 212-229): A 2% discount rate is used for the “Human Health” category and all GWP categories in accordance with the EPA’s SCC calculation from the Office of Management and Budget guidance²² and standard practice on social discounting.²³

In-text Citations:

22. US EPA. *Supplementary Material for the Regulatory Impact Analysis for the Final Rulemaking, “Standards of Performance for New, Reconstructed, and Modified Sources and Emissions Guidelines for Existing Sources: Oil and Natural Gas Sector Climate Review” - EPA Report on the Social Cost of Greenhouse Gases: Estimates Incorporating Recent Scientific Advances.* (2023).
23. Drupp, M. A., Freeman, M. C., Groom, B. & Nesje, F. Discounting Disentangled. *American Economic Journal: Economic Policy* **10**, 109–134 (2018).

8. *It is not clear why the authors use average of MYs 2010-2018 to characterize a MY 2024 diesel vehicle. It seems to me that a relevant comparison should be of a new EV relative to a new diesel.*

Response: We appreciate the reviewer raising this point. Our choice of emission factors are based on a lot of background research we did as part of the modeling in Tong et al.² We consulted with experts in measurement of emissions specifically from heavy-duty diesel trucks to ensure our emission factors were representative. Tong et al.² demonstrates that beyond MY 2010–2018 trucks with DPF and SCR, only negligible changes to air pollution emissions can be expected due to the physical and chemical constraints of the technology even when performing perfectly. So we would not expect big changes in emissions, absent some unforeseen technological breakthrough (and regulatory policy acting as a forcing function). It is true that we could, theoretically, establish anticipated emission factors for each new model year but these would be based on emissions standards, as opposed to measured values and we wanted our emission factors to be grounded in measured values because of the outsized impact that superemitters can have (due to malfunctioning emissions controls, etc.) For these two reasons (minimal improvements beyond MY 2010-2018, and lack of fleet-wide representative emissions for new trucks in 2025 and beyond), we opted to use the MY 2010-2018 data. To address this ambiguity, we have added justification of this modeling decision in lines 742 through 749.

Revised Text (Lines 746-753): For external costs in 2024, we assume emission factors match those of model year 2010 through 2018 diesel class 8 trucks due to the physical and chemical constraints of the pollution diesel particulate filters and selective catalytic reduction that would limit significant changes in EFs by year absent any disruptive change in control technologies. Following this logic, we assume only incremental improvements to 2024 emission factors when extrapolating to emissions and external costs in 2035, as per Table S8. These assumptions align with Tong et al.²² and avoid reliance on arbitrary assumptions about emission standards, policies, and their impacts on real-world vehicle performance.

In-text Citations

22. Tong, F., Jenn, A., Wolfson, D., Scown, C. D. & Auffhammer, M. Health and Climate Impacts from Long-Haul Truck Electrification. *Environ. Sci. Technol.* **55**, 8514–8523 (2021).

9. *The paper should mention some of the key limitations at the end*

Response: We've added a "Limitations" section in lines 903 through 938. We thank the reviewer for suggesting this improvement.

Revised Text (Lines 905-940):

Limitations

Throughout this study there are several sources of uncertainty and simplifications that future analyses should consider when further modeling impacts associated with electrification of HDVs. While drive cycles are an acceptable proxy for vehicle behavior, they are not fully representative of real-world driving decisions and may mischaracterize the energy efficiency of HDVs. The impacts of regenerative braking are highly dependent on driver behavior and road conditions that are difficult to capture in drive cycles as well. We provide Fig. S46 to explore how our results vary under a simplified drive cycle with no regenerative braking savings. Additionally, our study employs a simple battery performance and degradation model that estimates cycling and calendar aging, but factors like regional temperature, driver behavior, road conditions, local traffic flow, etc. could all impact degradation and instantaneous performance. Improved modeling of these factors along with a physio-chemical model of battery performance may improve results but at the cost of increased computational demand.

The ISRM used to model air pollution impacts in this study is also designed for computational efficiency but it makes several simplifying assumptions, as discussed in greater detail in McNeil

et al.⁷ and Tessum et al.²⁴ Additionally, the ISRM relies on background concentration and demographic data that is not updated to the current year or future years. Updating this data may improve results but is a tremendous undertaking for a study not primarily focused on background concentrations and demographics; thus it is not standard practice throughout literature.^{7,21,78,79} Development of standard scenarios for future demographic changes and background air pollutant concentrations would be of great value to the research community.

Future charging prices for long-haul HDVs is also highly uncertain and would require dedicated scenario development, considering the current lack of infrastructure for this EV use case, regional and temporal variability in electricity prices, and trends in long-term electricity portfolios. Our sensitivity analysis on charging and diesel prices aims to mitigate this uncertainty but may not fully capture possible outcomes. Figure S14 and S15 further visualize the impacts of variable charging prices on our final results. Additionally, while large scale changes to our electricity infrastructure that would substantially change marginal generation are unlikely, they are not impossible and could impact external costs associated with BE-HDVs. Finally, variations in the capital and operational expenditures of diesel and BE-HDV's could impact private costs of long-haul freight explored in this study. In particular, future prices of Li-ion batteries are highly uncertain due to their reliance on critical materials, unstable international market dynamics, and their potential for technological improvements.

Additional Figures:

Fig. S46. Social costs with a simplified drive cycle (constant 65 mph).

Fig. S14. Social costs under a low renewable cost scenario for a variety of years and electricity prices with a 2% discount rate.

Fig. S15. Social costs under a high renewable cost scenario for a variety of years and electricity prices with a 2% discount rate.

In-text Citations:

7. McNeil, W. H. *et al.* Impact of truck electrification on air pollution disparities in the United States. *Nat. Sustain.* (2025) doi:10.1038/s41893-025-01515-x.
24. Tessum, C. W. *et al.* Inequity in consumption of goods and services adds to racial-ethnic disparities in air pollution exposure. *Proc Natl Acad Sci USA* **116**, 6001–6006 (2019).

21. McNeil, W. H., Tong, F., Harley, R. A., Auffhammer, M. & Scown, C. D. Corridor-Level Impacts of Battery-Electric Heavy-Duty Trucks and the Effects of Policy in the United States. *Environ. Sci. Technol.* **58**, 33–42 (2024).
78. Goforth, T. & Nock, D. Air pollution disparities and equality assessments of US national decarbonization strategies. *Nat. Commun.* **13**, 7488 (2022).
79. Jackson, C. M., Holloway, T. & Tessum, C. W. City-scale analysis of annual ambient PM_{2.5} source contributions with the InMAP reduced-complexity air quality model: a case study of Madison, Wisconsin. *Environmental Research: Infrastructure and Sustainability* **3**, 015002 (2023).

10. The authors should check the y axis labels in Figs 2B-2C – it seems they probably mean USD per gallon. And in Fig 4A-4B is ng/m³ of PM_{2.5} correct ? (or is it ug/m³)

Response: We appreciate the reviewer’s thorough reading of the figures and have corrected the typo in Figure 2B-2C; units for diesel prices are indeed in USD per gallon. The units in Figure 4A and 4B are correct as ng/m³.

Reviewer #2 Response References:

1. Jenn, A., Clark-Sutton, K., Gallaher, M. & Petrusa, J. Environmental impacts of extreme fast charging. *Environmental Research Letters* **15**, 094060 (2020).
2. Tong, F., Jenn, A., Wolfson, D., Scown, C. D. & Auffhammer, M. Health and Climate Impacts from Long-Haul Truck Electrification. *Environ. Sci. Technol.* **55**, 8514–8523 (2021).
3. U.S. EPA. *Overview of EPA’s MOfor Vehicle Emission Simulator (MOVES4)*. (U.S.EPA, 2023).
4. Cai, H., Burnham, A., Wang, M., Hang, W. & Vyas, A. *The GREET Model Expansion for Well-to-Wheels Analysis of Heavy-Duty Vehicles*. (Argonne National Laboratory, 2015).
5. Gagnon, P. & Cole, W. Planning for the evolution of the electric grid with a long-run marginal emission rate. *iScience* **25**,103915 (2022).

6. Siler-Evans, K., Azevedo, I. L. & Morgan, M. G. Marginal emissions factors for the U.S. electricity system. *Environ. Sci. Technol.* **46**, 4742–4748 (2012).
7. Graff Zivin, J. S., Kotchen, M. J. & Mansur, E. T. Spatial and temporal heterogeneity of marginal emissions: Implications for electric cars and other electricity-shifting policies. *J. Econ. Behav. Organ.* **107**, 248–268 (2014).
8. Archsmith, J., Kendall, A. & Rapson, D. From cradle to junkyard: assessing the life cycle greenhouse gas benefits of electric vehicles. *Research in Transportation Economics* **52**, 72–90 (2015).
9. Hoehne, C. G. & Chester, M. V. Optimizing plug-in electric vehicle and vehicle-to-grid charge scheduling to minimize carbon emissions. *Energy* **115**, 646–657 (2016).
10. McNeil, W. H., Tong, F., Harley, R. A., Auffhammer, M. & Scown, C. D. Corridor-Level Impacts of Battery-Electric Heavy-Duty Trucks and the Effects of Policy in the United States. *Environ. Sci. Technol.* **58**, 33–42 (2024).
11. McNeil, W. H., Porzio, J., Tong, F., Harley, R. A., Auffhammer, M. & Scown, C. D. Impact of truck electrification on air pollution disparities in the United States. *Nat. Sustain.* (2025). doi:10.1038/s41893-025-01515-x
12. Ryan, N. A., Johnson, J. X. & Keoleian, G. A. Comparative assessment of models and methods to calculate grid electricity emissions. *Environ. Sci. Technol.* **50**, 8937–8953 (2016).
13. Holland, S. P., Kotchen, M. J., Mansur, E. T. & Yates, A. J. Why marginal CO₂ emissions are not decreasing for US electricity: Estimates and implications for climate policy. *Proc Natl Acad Sci USA* **119**, (2022).
14. Bruchon, M., Chen, Z. L. & Michalek, J. Cleaning up while Changing Gears: The Role of Battery Design, Fossil Fuel Power Plants, and Vehicle Policy for Reducing Emissions in the Transition to Electric Vehicles. *Environ. Sci. Technol.* **58**, 3787–3799 (2024).
15. Burnham, A., Gohlke, D., Rush, L., Stephens, T., Zhou, Y., Delucchi, M., Birky, A., Hunter, C., Lin, Z., Ou, S., Xie, F., Proctor, C., Wiryadinata, S., Liu, N. & Bloor, M. *Comprehensive Total Cost of Ownership Quantification for Vehicles with Different Size Classes and Powertrains*. (Argonne National Laboratory (ANL), 2021). doi:10.2172/1780970
16. Wall, J. Interview with John Wall, Former CTO of Cummins. (2023).

17. Hunter, C., Penev, M., Reznicek, E., Lustbader, J., Birky, A. & Zhang, C. *Spatial and Temporal Analysis of the Total Cost of Ownership for Class 8 Tractors and Class 4 Parcel Delivery Trucks*. (National Renewable Energy Laboratory, 2021).
18. Goforth, T. & Nock, D. Air pollution disparities and equality assessments of US national decarbonization strategies. *Nat. Commun.* **13**, 7488 (2022).
19. Jackson, C. M., Holloway, T. & Tessum, C. W. City-scale analysis of annual ambient PM_{2.5} source contributions with the InMAP reduced-complexity air quality model: a case study of Madison, Wisconsin. *Environmental Research: Infrastructure and Sustainability* **3**, 015002 (2023).
20. Cohon, J. L., Cropper, M. L., Cullen, M. R., English, M. R., Field, C. B., Greenbaum, D. S., Hammitt, J. K., Henderson, R. F., Kling, C. L., Krupnick, A. J., Lee, R., Matthews, H. S., McKone, T. E., Metcalf, G. E., Newell, R. G., Revesz, R. L., Wing, I. S. & Surles, T. G. *Hidden costs of energy: unpriced consequences of energy production and use*. (National Academies Press, 2010). doi:10.17226/12794
21. Drupp, M. A., Freeman, M. C., Groom, B. & Nesje, F. Discounting Disentangled. *American Economic Journal: Economic Policy* **10**, 109–134 (2018).

Reviewer #2:

We thank you for your excellent comments on our paper. We have taken your suggestions seriously and have done our best to address them in the manuscript. Doing so has significantly improved the manuscript. Our point-by-point responses and descriptions of the changes are included below.

1. In their rebuttal, the authors confirm that their large externality values for electricity generation are a consequence of applying a marginal model. However, they do not disclose the emission rates involved. Without having an idea of the emission rates, it is not possible to understand the authors' results. Both users of their results to inform policies and other researchers building on their study aiming refining their model would need to have a notion of the emission rates involved to be able to compare and reconcile any possible differences in findings. While I understand that emission rates are drastically different for each EGU, a summary table of the resulting average emission rates per kWh (the average across all EGUs meeting the BE-HDV demand) would still be quite useful to the readers, and even the paper by Jenn et al. cited by the authors has published average emission rates under different scenarios.

Response:

We thank the reviewer for highlighting this point. We have added Table S12 and S13 providing the average US electricity generation marginal emission factors excluding solar and wind generation by scenario and the average US marginal emissions per total electricity demand from battery charging by scenario respectively. We direct readers to this Table in Lines 106-109.

Revised Lines (106-109): Table S12 provides average US electricity generation marginal emission factors excluding solar and wind generation. Table S13 provides the average US marginal emissions per total electricity demand from battery charging by scenario.

Revised Table S12:

Table S12. Average US electricity generation marginal emission factors, excluding solar and wind generators, by scenario.

Scenario	NO _x (g/kWh)	SO ₂ (g/kWh)	N ₂ O (g/kWh)	CH ₄ (g/kWh)	PM _{2.5} (g/kWh)	CO ₂ (kg/kWh)
LowRE 2025	0.4350	0.7553	0.0140	0.0998	0.1110	0.4786
LowRE 2035	0.2554	0.3337	0.0074	0.0534	0.0629	0.2632
HighRE 2025	0.5081	0.8281	0.0155	0.1101	0.1151	0.5824
HighRE 2035	0.4153	0.5933	0.0133	0.0955	0.1085	0.4383

Revised Table S13:**Table S13.** Average US marginal emissions per total electricity demand from battery charging by scenario.

Scenario	NO _x (g/kWh)	SO ₂ (g/kWh)	N ₂ O (g/kWh)	CH ₄ (g/kWh)	PM _{2.5} (g/kWh)	CO ₂ (kg/kWh)
LowRE 2025	0.4113	0.7142	0.0133	0.0944	0.1049	0.4526
LowRE 2035	0.1945	0.2541	0.0056	0.0407	0.0479	0.2004
HighRE 2025	0.4932	0.8038	0.0151	0.1079	0.1181	0.5654
HighRE 2035	0.3686	0.5264	0.0118	0.0848	0.0962	0.3889

2. The authors need to describe in a little bit more detail the “marginal” scenario assumed for electricity. The authors mention in line 630 that they simulate 100 trucks. There are more 3-4 million class 8 trucks alone in the US (e.g. <https://www.bts.gov/browse-statistical-products-and-data/surveys/vius/vehicle-stats-state-vehicle-type-and-gywr>). If this entire truck fleet were to be electrified, they would increase total US electricity consumption by on the order of 10% (~2 kWh/miles x ~200 billion VMT/year[1] = ~4e11 kWh).

[1] <https://www.bts.gov/browse-statistical-products-and-data/freight-facts-and-figures/vehicle-miles-traveled-highway>)

While electrifying 3-4 million trucks would not be realistic, especially in a short time frame, any realistic policy scenario would also involve more than 100 trucks. Therefore it raises the question as to whether the authors calculated the short run marginal to meet the additional demand of just 100 trucks – which probably would not be relevant to any realistic policy scenario (Especially as far as results for the future/2035, when it is not clear that meaningful adoption of electric HDVs would be met with existing electricity generating units/current short-run marginal).

Response:

We appreciate the reviewer raising this concern. The results in this study represent the costs associated with a marginal long-haul truck, i.e. a single or small quantity of trucks introduced to the infrastructure present in the United States. Our goal in running a Markov Chain Monte Carlo with 100 trucks is simply to estimate the impact per VMT at the margin, as opposed to *only* conveying the impacts of electrifying exactly 100 trucks. We settled on simulating 100 trucks purely due to computational limitations because of the computational resources and time it takes to run our model iteratively. The grid portion of our model defines short-run marginal generators for each region and each hour of the year across different scenarios, but it does not specify exactly how much additional load must be added before this no longer holds true, and a

different type of generator would be on the margin. This is a fundamental limitation of the Cambium data we rely on from the NREL scenarios. In other words, our results (which are inherently linear, as they are presented in terms of damages per VMT) should be representative of electrification well beyond 100 trucks (likely for several orders of magnitude larger fleets), but we cannot say with certainty the maximum scale beyond which our per-VMT results would no longer be representative of realistic grid responses and emissions. As the reviewer notes, full electrification of the 3-4 million class 8 trucks would not occur instantaneously and thus a different modeling approach would be more appropriate to capture the interdependence of large-scale fleet electrification and long-term grid planning.

Regarding the reviewer's note about future results (e.g. 2035), we *do* want to clarify that the marginal generators are tied to future scenarios and what types of generators are expected to meet marginal loads in those years (*see existing Line 564*). In other words, the 2035 scenario includes marginal generator types that are expected to meet loads in 2035; they are not based on the current short-run marginal generator types. However, we do use the current locations and emission factors associated with each type of generator in each region because we do not have a defensible way of predicting where new power plants may be constructed, or which current generators will be decommissioned.

We appreciate that our manuscript could be clearer about why we simulate 100 trucks and how our results should be interpreted. To address this, we have revised Lines 76-78 and 96-99, which further directs readers to Discussion S2. We also make further changes along these lines, as outlined in our response to the reviewer's next comment (see below). Additionally, we add discussion on the inherent limitations of using a short-run marginal generation model in Lines 725-732. Finally, we further reiterate that all future grid scenarios account for expected changes in marginal generation based on NREL's Cambium datasets in Lines 565-567.

Revised Text (Lines 76-78): Specifically, we answer whether the electrification of a marginal long-haul HDVs with Li-ion batteries reduces GHG emissions *and* decreases the burden on human health, and how the value of those impacts compares to the private costs for truck operators.

Revised Text (Lines 96-99): We estimate the private costs of marginal Li-ion BE-HDVs and diesel HDVs by modeling the annual activity of a fleet of 100 HDVs performing long-haul trips via a Markov Chain Monte Carlo simulation and determining their average net present value (NPV) over their four-year average lifetime.¹⁴⁻¹⁶ Further elaboration on this modeling decision is provided in Discussion S2.

Revised Discussion S2:

Discussion S2

Our Markov Chain Monte Carlo performs 100 runs (i.e. simulates a fleet of 100 trucks) in order to estimate the average marginal impact of HDV electrification with Li-ion batteries per VMT. This fleet size is selected purely due to computational limitations and the time it takes to run our model iteratively. Additionally, the short-run marginal grid model we employ defines marginal generators for each region and each hour of the year across different scenarios, but it does not specify exactly how much additional capacity is available at each generator before a different type of generator would be called upon. This is a fundamental limitation of the Cambium data we rely on from the NREL scenarios. However, given the relatively small instantaneous demand from charging a truck (on the order of magnitude of tens to hundreds of kW), and the wide dispersal of vehicles across the US, our results should be representative of long-haul HDV electrification at much larger scales than 100 trucks (likely for fleets several orders of magnitude larger). Nevertheless, we cannot say with certainty the maximum scale beyond which our inherently linear per-VMT results would no longer be representative of realistic grid responses and emissions.

Revised Text (Lines 714-719): A final limitation in our energy and emissions modeling is our use of short-run marginal generators. Our per-VMT emissions and air pollution results can be used for comparatively small increases in grid loads, but marginal generators may change for larger changes in regional loads. We cannot say with certainty the maximum scale beyond which our per-VMT results would no longer be representative of realistic grid responses and emissions.

Revised Text (Lines 565-567): Specifically, this grid model uses NREL's Cambium datasets to estimate how hourly marginal generation will change by region into the future.

2.1. The fact that all of the authors results are based on simulating just 100 trucks should be clarified earlier in the manuscript. Without that information it is difficult, for example, to make sense of the ~0.01 ng/m³ change in PM_{2.5} in figures 4A/4B (which is only clear once one realizes that this is due to just 100 trucks).

Response:

We agree with this point and direct the reviewer to our revision presented in the previous comment. Additionally, we revised the accompanying text for Fig. 3 (although the reviewer

references Fig 4, we think the intention was to refer to Fig 3, which shows changes in PM_{2.5} concentrations) (Lines 225-228) to provide further clarity.

Revised Text (Lines 225-228): Fig. 3. External costs of long-haul BE-HDVs and diesel HDVs in varying years and renewable energy cost scenarios. (A) Change in PM_{2.5} from the simulated marginal fleet of 100 diesel HDVs in 2025. (B) Change in PM_{2.5} from the simulated marginal fleet of 100 BE-HDVs in 2025 under a low renewable energy cost scenario.

3. I appreciate that the authors added a limitations section, but I think it that (i) also needs to mention that there is substantial uncertainty about both the VSL and the magnitude of the mortality effects of ambient PM_{2.5} exposure; and (ii) because the uncertainty is not fully captured by the confidence intervals given by the authors, it seems to me that it would be appropriate to list the key limitations in the main text/discussion, as opposed to relegating all of it to the end of the methods section.

Response:

We thank the reviewer for providing this feedback. With regards to (ii), we have added Lines 729-731 in the Limitations to address these points. With regards to (i), we have directed readers to our *Limitation* section in the *Discussion* of our manuscript (Lines 329-330) in order to preserve the focus of section while also providing clear direction to the key limitations of our methods. Additionally, we add further mention of these limitations throughout the main text (Lines 200-206 and Lines 280-285) to make sure that discussion of the limitations is not solely relegated to the Methods.

Revised Text (Lines 729-731): Beyond potential ISRM updates, a more thorough exploration of uncertainty regarding the VSL and the magnitude of the mortality effects of ambient PM_{2.5} exposure would improve the quality of similar studies in the future.

Revised Text (Lines 329-330): Additionally, limitations of our methodology can be found in the *Limitations* section.

Revised Text (Lines 200-206): Even for impacts within the US, it is worth noting that the translation to monetized damages through the use of SCC and value of a statistical life (VSL) come with substantial uncertainty, and a full exploration of this uncertainty warrants additional studies dedicated to this topic. We apply these costs in a consistent manner across BE-HDV and diesel HDV scenarios to ensure a fair comparison. However, uncertainty associated with the SCC and VSL can impact the relative importance of private and external costs in the total social cost calculations discussed in the following section.

Revised Text (Lines 280-285): As noted in the previous section, the VSL and SCC values introduce additional uncertainty beyond what is shown in the uncertainty bars for this study and a more in-depth exploration of this uncertainty warrants its own study. Variations in the SCC and VSL do impact their importance in the context of total social costs. A more in-depth comparison of our values to literature is provided in the Methods.